# EMPOWERING LLM TOOL INVOCATION WITH TOOL-CALL REWARD MODEL

**Da Ma**[1]**, Ziyue Yang**[1]**, Hongshen Xu**[1]**, Haotian Fang**[1]**, Kai Yu**[1,3,4]**, Lu Chen**[1,2,3,4*]
[1]X-LANCE Lab, School of Computer Science, Shanghai Jiao Tong University, Shanghai, China
[2]Shanghai Innovation Institution, Shanghai, China
[3]Jiangsu Key Lab of Language Computing, Suzhou, China
[4]Suzhou Laboratory, Suzhou, China
{mada123, chenlusz}@sjtu.edu.cn

## ABSTRACT

Large Language Models (LLMs) have recently alleviated limitations in outdated internal knowledge and computational inaccuracies by invoking external tools such as search engines and code generation. While reinforcement learning (RL) has substantially enhanced tool usage in LLMs, most existing agentic RL approaches rely solely on outcome-only reward signals, which assign credit at a coarse granularity and often induce gradient conflict (e.g., correct tool calls may be penalized due to incorrect final answers). To address this, we propose the *Tool-call Reward Model* (TRM), a specialized process reward model meticulously designed to evaluate and reward each tool invocation. Since previous PRM research has predominantly focused on traditional reasoning tasks such as step-wise mathematical reasoning, the introduction of TRM brings two unique challenges: (1) limited understanding of how to construct effective TRMs, including data requirements and model size; and (2) difficulties integrating TRM with classical RL algorithms such as PPO and GRPO, where naive adaptation may lead to reward hacking (minimizing tool calls to avoid penalties). To tackle these challenges, we establish a systematic TRM construction workflow and propose refined credit assignment and turn-level advantage estimation for effective integration with PPO and GRPO. Experiments show that a 3B TRM trained on 10K samples achieves robust performance. On search-based QA and Python code-based math tasks, integrating TRM consistently outperforms outcome-only reward RL methods across models of different sizes.[1]

## 1 INTRODUCTION

Large Language Models (LLMs) have demonstrated sophisticated proficiency in addressing complex tasks, profoundly impacting a broad spectrum of domains (OpenAI, 2023; Guo et al., 2025; Yang et al., 2025). However, LLMs are fundamentally limited by the static nature of their internal knowledge and their propensity to make computational errors (Schick et al., 2023; Qian et al., 2025a). To overcome these challenges, LLMs increasingly invoke external tools, such as search engines for accessing up-to-date information (Jin et al., 2025; Chen et al., 2025b) and code generation for solving complex mathematical problems (Liao et al., 2024; Feng et al., 2025).

With tool invocation playing an increasingly important role in overcoming LLM limitations, reinforcement learning (RL), proven effective in traditional reasoning tasks (Guo et al., 2025; Team, 2025; Team et al., 2025; Wang et al., 2024), has been widely used to enhance tool usage. In practice, most RL-based approaches (Jin et al., 2025; Song et al., 2025; Feng et al., 2025; Li et al., 2025b) for tool invocation rely solely on outcome reward signals, evaluating only the correctness of the final output (e.g., math answer correctness) while overlooking the quality of intermediate tool calls. Consequently, credit for each tool call in a trajectory is assigned solely based on the final outcome,

---

*Corresponding author
[1]Avaliable on `https://github.com/OpenDFM/TRM`

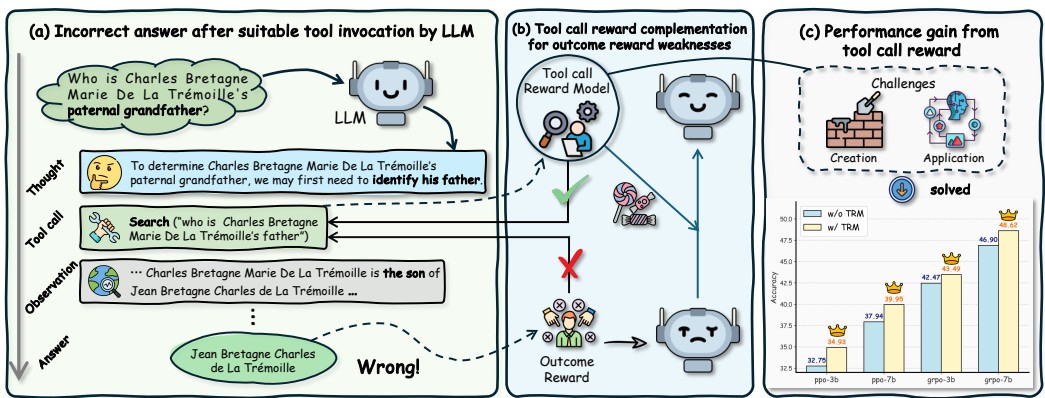

Figure 1: Overview of TRM for improving tool invocation in LLMs: (a) limitations of outcome-only reward, (b) benefits of tool call reward, and (c) performance gains from integrating tool call reward.

irrespective of its individual quality or usefulness. With uniform treatment of tool calls, this approach limits the ability of the model to learn effective tool usage, potentially resulting in unstable or suboptimal performance. For example, if the final answer is incorrect, a trajectory with correct intermediate tool usage is still penalized (Figure 1-a and Figure 1-b)[2]. This discourages learning of effective tool invocation strategies and causes *gradient conflict* (Lightman et al., 2024; Deng et al., 2025b), often leading to unstable tool usage and degraded performance.

To this end, we propose the *Tool-call Reward Model* (TRM), which quantitatively measures the utility of each tool invocation and assigns corresponding rewards. Although TRM can be viewed as a specific type of process reward model (PRM), prior PRM research (Lightman et al., 2024; Zhang et al., 2025b; Setlur et al., 2025) has predominantly focused on traditional reasoning tasks, leaving tool invocation underexplored. TRM fills this gap by enabling fine-grained monitoring of tool usage, thereby facilitating more appropriate tool invocation (Figure 1-b). However, introducing TRM raises two new key challenges (Figure 1-c): 1) *TRM creation*: how to construct an effective TRM, and 2) *TRM application*: how to integrate it with classical RL algorithms.

For the first challenge, the main difficulty lies in the limited understanding of TRMs, including how to construct training data, the required data volume, and the suitable model size. To address this, we develop a workflow to distill training data from frontier LLMs (§ 2.2) and systematically analyze the impact of data volume and model size on TRM performance (§ 3.1). Beyond this, integrating TRM with classical RL algorithms such as PPO Schulman et al. (2017) and GRPO Shao et al. (2024) remains an open challenge, as directly transferring approaches that combine standard PRM and RL algorithms may not work well for TRM. For instance, in GRPO, our experiments demonstrate that group-level advantage estimation (Shao et al., 2024) of tool call reward can result in reward hacking, where the model prefers fewer tool calls over effective usage (see Appendix E.1). To address these issues, we refine the credit assignment strategy by allocating tool call rewards to the end of each tool invocation, and introduce turn-level advantage estimation in GRPO (§ 2.3). Ultimately, our experiments show that the proposed methods yield better overall model performance (Figure 1-c, § 3.2). Furthermore, we observe that TRM enhances generalization in tool invocation, enabling the model to flexibly adapt to unseen tools (§ 3.3).

In summary, this work makes the following three contributions:1) We propose the *Tool-call Reward Model* (TRM) and conduct a thorough investigation into its construction. 2) We develop and analyze new algorithms for integrating TRM with classical RL methods, including refined credit assignment strategies (PPO) and step-wise advantage estimation (GRPO). 3) We validate our approaches through extensive experiments, demonstrating significant improvements in model performance. We plan to make our data and code publicly available to facilitate future research.

---

[2]A reasonable way to determine the paternal grandfather of a person is to first determine the father of the person, then the father of that father.

## 2 METHODOLOGY

We introduce a Tool-call Reward Model (TRM) to resolve gradient conflict from outcome-only rewards by supplying fine-grained, per-call utility signals that stabilize the tool invocation. In this section, we (i) formalize the multi-turn RL framework for tool invocation in LLMs, (ii) detail the construction of TRM, including training data distillation and model optimization, and (iii) integrate TRM with classic RL algorithms by proposing turn-level credit assignment and enhancing GRPO with turn-level advantage estimation.

### 2.1 PROBLEM FORMULATION

We formalize multi-turn tool invocation in LLMs as a sequential decision-making process under the reinforcement learning framework. Following the ReAct paradigm (Yao et al., 2023), the LLM alternates between reasoning steps and tool invocations, enabling dynamic planning and external information gathering for more robust and interpretable task-solving. Formally, consider a prompt $p$ and an LLM $\pi$ parameterized by $\theta$. Given $p$, the LLM $\pi$ engages in multiple rounds of tool invocation, where at each round, the model reasons over the current information and decides on the next tool action. This iterative process continues until the model is ready to produce the final answer. Finally, the LLM $\pi$ generates a trajectory

$$\tau = (p, t_1, a_1, o_1, \ldots, t_{n_\tau}, a_{n_\tau}, o_{n_\tau}, t_{n_\tau+1}, y), \tag{1}$$

where $t_i$ $(1 \leq i \leq n_\tau + 1)$ denotes the reasoning thought, $a_i$ and $o_i$ $(1 \leq i \leq n_\tau)$ is the tool invoked and its corresponding output at turn $i$, $n_\tau$ is the total number of tool invocation rounds, and $y$ is the final answer produced by the LLM $\pi$. Here, we refer to each triplet $(t_i, a_i, o_i)$ as a single *turn* in the interaction.[3]

Given this formulation, our objective is to optimize the policy $\pi_\theta$ to maximize the likelihood of producing the correct final answer $y$ at the end of the trajectory. Formally, the learning objective is to maximize the expected correctness of the final answer $y$ over trajectories generated by the policy $\pi_\theta$:

$$\max_\theta \mathbb{E}_{\tau \sim \pi_\theta} \left[ \mathbb{I}\left(y = y^*\right) \right], \tag{2}$$

where $y^*$ is the ground-truth answer and $\mathbb{I}\left(\cdot\right)$ is the indicator function.

### 2.2 CONSTRUCTION OF TRM

**Data Distillation** We first describe the process of distilling high-quality training data for TRM from frontier LLMs (Figure 2-a). This process consists of two main steps: 1) *rollout collection* and 2) *tool call evaluation*. In the rollout collection step, the model is provided with a set of prompts and a tool-enabled environment, and generates multi-turn trajectories by autonomously invoking tools to complete the task. For each collected rollout, we further evaluate every tool call $a_i$ by re-feeding the whole trajectory into the model to assess its utility. Specifically, we assign two binary scores for each tool call $a_i$:

- *necessity* $s_{\text{ne}}^i$: whether the tool call contributes substantive progress toward task completion
- *quality* $s_{\text{q}}^i$: whether the tool is invoked with reasonable parameters or used correctly

Hence, a tool call is assigned a score of $1$ only when it is both necessary for task progress and executed with high quality; if either criterion is not met, the score is $0$. Formally, for a tool call $a_i$, the final score is defined as:

$$s^i = s_{\text{ne}}^i \cdot s_{\text{q}}^i, \tag{3}$$

where $s_{\text{ne}}^i, s_{\text{q}}^i \in \{0, 1\}$. The detailed design of prompts are illustrated in Appendix A.1.

---

[3]The final turn consists of both reasoning and the generation of the final answer, without involving any tool call.

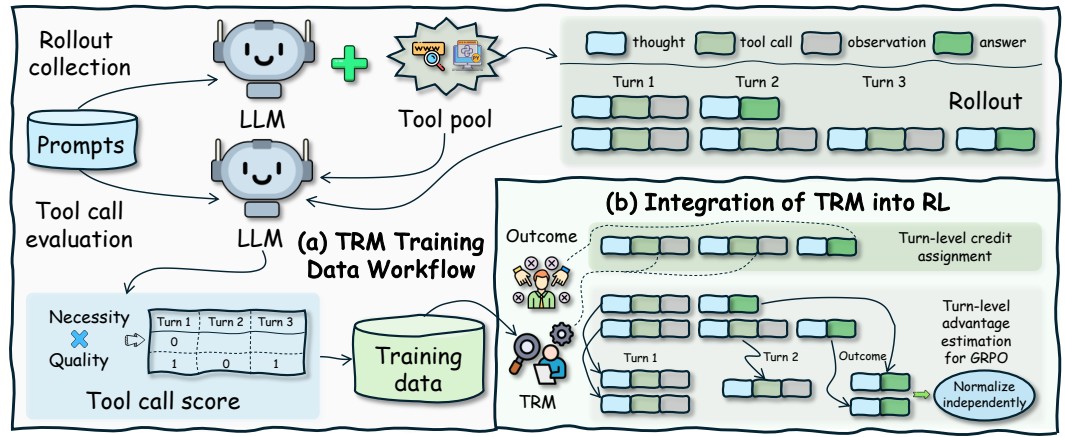

Figure 2: TRM-guided LLM tool invocation. (a) Generation of tool invocation trajectories and turn-level utility labels for TRM training. (b) Turn-level credit assignment and GRPO adaptation via turn-level advantage estimation.

**TRM Training**   The TRM adopts a transformer-based (Vaswani et al., 2017) LLM as its backbone. To adapt the model for tool-call utility prediction, we replace the original language modeling head (used for next-token prediction) with a binary classification head consisting of a single linear layer. Specifically, for each tool call $a_i$, the model produces a probability $\tilde{s}^i \in [0, 1]$ based on the hidden state of the last token of the tool call output $o_i$. This score indicates the predicted utility of the tool call. During training, the TRM is optimized using a binary cross-entropy loss[4]:

$$\mathcal{L}_{\text{BCE}} = \mathbb{E}_\tau \left[ -\frac{1}{n_\tau} \sum_{i=1}^{n_\tau} \left( s^i \log \tilde{s}^i + \left(1 - s^i\right) \log \left(1 - \tilde{s}^i\right) \right) \right]. \tag{4}$$

## 2.3   INTEGRATION OF TRM WITH RL

With TRM in place, we proceed to integrate it into established RL algorithms to optimize tool invocation in LLMs. Specifically, we focus on two representative policy optimization methods[5]: Proximal Policy Optimization (PPO) and Group Relative Policy Optimization (GRPO).

**Turn-level Credit Assignment**   To achieve appropriate credit assignment throughout the trajectory, we combine TRM scores for intermediate tool invocations with the outcome reward for the final answer (Figure 2-b). In particular, for each turn $i$ $(1 \le i \le n_\tau)$ of trajectory $\tau$, the reward is given by the TRM score $\tilde{s}^i$, and for the final reasoning step $(i = n_\tau + 1)$, the reward is determined by the correctness of the final answer. Mathematically, the turn-level reward $\tilde{r}^i$ is defined as

$$\tilde{r}^i = \begin{cases} \tilde{s}^i, & 1 \le i \le n_\tau \\ \mathbb{I}(y = y^*), & i = n_\tau + 1 \end{cases}. \tag{5}$$

Both PPO and GRPO perform policy optimization at the token level, whereas our reward signals are defined at the turn level. To bridge this granularity gap, we also represent each trajectory as a sequence of tokens, $\tau = (x_1, x_2, \ldots, x_L)$, where $x_j$ is the $j$-th token. For each turn $i$ $(1 \le i \le n_\tau)$, we identify $e_i$ as the index of the last token of the tool call $a_i$. The set $\mathcal{E} = \{e_1, \ldots, e_{n_\tau}\}$ thus marks all tool-call-ending tokens. We further define a mapping $\mathcal{I}(j)$ that returns the corresponding turn index for any $j \in \mathcal{E}$, and set $\mathcal{I}(L) = n_\tau + 1$ for the final answer. To specify which tokens participate in policy optimization, we define $\mathcal{M} \subseteq \{1, \ldots, L\}$ as the set of indices of thought, tool call, and answer tokens that are not masked during RL training. These notations facilitate our subsequent discussion on the integration of TRM with RL.

---

[4]In practice, a score is also produced at the last token of the answer to indicate its correctness.

[5]For clarity, KL regularization is omitted in our discussion.

**Integration with PPO** To enable token-level policy optimization, we map turn-level rewards to the corresponding tokens by assigning the reward for each tool call to the last token of the associated action, and the outcome reward to the last token of the answer. Formally, the reward $r^j$ of token $x_j$ $(1 \le j \le L)$ is defined as

$$r^j = \begin{cases} \alpha \cdot \tilde{r}^{\mathcal{I}(j)}, & j \in \mathcal{E} \\ \tilde{r}^{\mathcal{I}(j)}, & j = L \\ 0, & \text{otherwise} \end{cases}, \tag{6}$$

where $\alpha \in (0, 1]$ is a hyperparameter controlling the weight of the TRM score. Advantage $A^j$ is then computed from $r^j$ (e.g., Generalized Advantage Estimation (Schulman et al., 2016)). With this token-level advantage, the PPO objective is formulated as

$$\mathcal{L}_{\text{PPO}} = \mathbb{E}_{\tau \sim \pi_\theta} \left[ \frac{1}{|\mathcal{M}|} \sum_{j \in \mathcal{M}} \min \left( w^j(\theta) \cdot A^j, \text{clip} \left( w^j(\theta), 1 - \epsilon, 1 + \epsilon \right) A^j \right) \right], \tag{7}$$

where $w^j(\theta) = \frac{\pi_\theta(x_j|x_{<j})}{\pi_{\theta_{\text{old}}}(x_j|x_{<j})}$ and $\epsilon$ is the clipping parameter.

**Integration with GRPO** GRPO is a policy optimization method that compares and normalizes rewards across a batch of trajectories to increase training efficiency. In GRPO, a group refers to a batch of $G$ trajectories $\{\tau_1, \ldots, \tau_G\}$. For each trajectory $\tau_g$ $(1 \le g \le G)$, variables such as $n_{\tau_g}, \mathcal{E}_g$, $\mathcal{M}_g$, and other notations follow the same definitions as in previous sections, with the addition of the trajectory index $g$. Across the group, we collect the TRM rewards $\tilde{\mathcal{R}}_{\text{trm}}^i$ (for any turn $i$) and outcome rewards $\tilde{\mathcal{R}}_{\text{out}}$ via

$$\tilde{\mathcal{R}}_{\text{trm}}^i = \left\{ \tilde{r}_g^i \mid 1 \le g \le G, \ i \le n_{\tau_g} \right\}, \quad \tilde{\mathcal{R}}_{\text{out}} = \left\{ \tilde{r}_g^{n_{\tau_g}+1} \mid 1 \le g \le G \right\}. \tag{8}$$

We then perform turn-level advantage estimation, where rewards for each turn are normalized independently across trajectories (Figure 2-b). In detail, for each turn $i$ and trajectory $\tau_g$, the normalized rewards are computed as

$$\hat{r}_g^i = \frac{\tilde{r}_g^i - \text{mean}\left(\tilde{\mathcal{R}}_{\text{trm}}^i\right)}{\text{std}\left(\tilde{\mathcal{R}}_{\text{trm}}^i\right)} \ (1 \le i \le n_{\tau_g}), \quad \hat{r}_g^{n_{\tau_g}+1} = \frac{\tilde{r}_g^{n_{\tau_g}+1} - \text{mean}\left(\tilde{\mathcal{R}}_{\text{out}}\right)}{\text{std}\left(\tilde{\mathcal{R}}_{\text{out}}\right)}. \tag{9}$$

These normalized rewards are then assigned to the corresponding tokens, and token-level advantages are computed via discounted aggregation:

$$r_g^j = \begin{cases} \alpha \cdot \hat{r}_g^{\mathcal{I}(j)}, & j \in \mathcal{E}_g \\ \hat{r}_g^{n_{\tau_g}+1}, & j = L_g \\ 0, & \text{otherwise} \end{cases}, \quad A_g^j = r_g^{L_g} + \sum_{m=j}^{L_g-1} \gamma^{m-j} r_g^m, \tag{10}$$

where $\alpha$ is a weighting hyperparameter and $\gamma$ is the discount factor[6]. With this token-level advantage, the GRPO objective is formulated as

$$\mathcal{L}_{\text{GRPO}} = \mathbb{E}_{\{\tau_g\} \sim \pi_\theta} \left[ \frac{1}{G} \sum_{g=1}^{G} \frac{1}{|\mathcal{M}|} \sum_{j \in \mathcal{M}} \min \left( w_g^j(\theta) \cdot A_g^j, \text{clip} \left( w_g^j(\theta), 1 - \epsilon, 1 + \epsilon \right) A_g^j \right) \right]. \tag{11}$$

## 3 EXPERIMENTS

In this section, we focus on two key aspects:

- *TRM exploration*: How can we obtain an effective TRM?
- *TRM exploitation*: Does introducing TRM improve the tool-use capabilities of LLMs?

---

[6]Masked tokens are skipped when computing the discounted sum of normalized rewards.

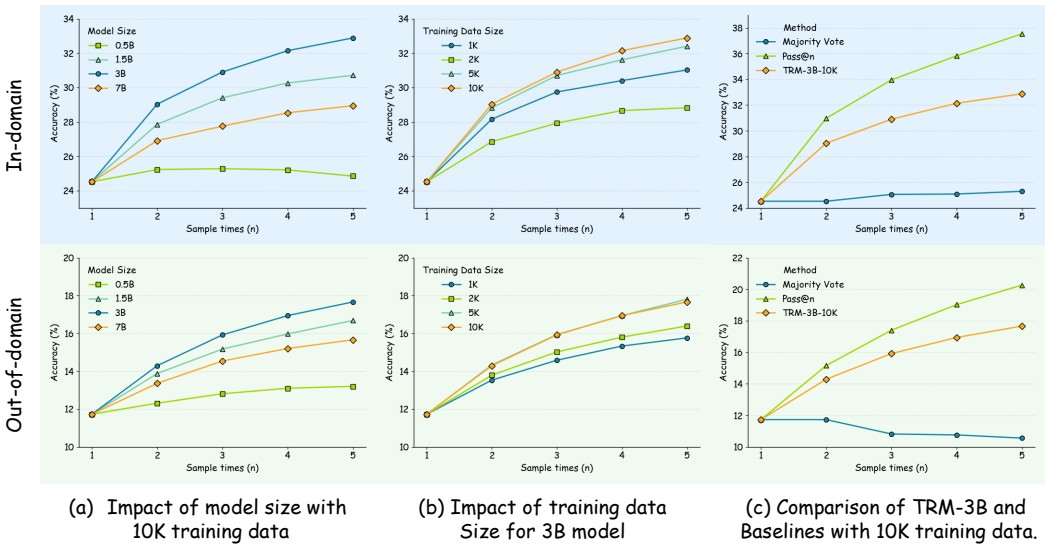

Figure 3: TRM performance comparison under different settings

## 3.1 EXPLORATION OF TRM

**Training Model and Data**    We use the Qwen2.5 (Qwen et al., 2025) series as the backbone architecture for TRM. For training data, we sample 15K prompts each from the HotpotQA (Yang et al., 2018) and NQ (Kočiský et al., 2018) training sets. Rollouts are generated using DeepSeek-R1 (Guo et al., 2025), which interacts with a search environment (Jin et al., 2025) to produce multi-turn trajectories. Each trajectory is annotated with turn-level utility labels based on necessity and quality by DeepSeek-R1. Finally, we randomly sample 10K labeled trajectories for TRM training. More training details are in Appendix B.1.

**Evaluation**    To evaluate TRM effectiveness, we use a best-of-n strategy (Lightman et al., 2024; Luo et al., 2025): for each prompt, $n$ candidate trajectories are generated, and the one with the highest TRM score is selected. The score for a trajectory $\tau$ is computed as the product of all tool call scores, i.e., $S(\tau) = \prod_{i=1}^{n_\tau+1} \tilde{s}^i$.[7] Evaluation is conducted in both in-domain (HotpotQA validation prompts) and out-of-domain (2WikiMultiHopQA (Ho et al., 2020) validation prompts) settings. All candidate trajectories are generated by the Search-R1 (Jin et al., 2025) model, which is PPO-trained based on Qwen2.5-7B. More details are in Appendix C.1.

**Results and Analysis**    According to the results in Figure 3, we observe following key trends:

> **Key Takeaways for TRM Exploration**
>
> - *Mid-sized TRMs (1.5B/3B) deliver optimal performance with 10K training samples,* while larger models (e.g., 7B) may be prone to overfitting given the same data scale.
> - *10K labeled trajectories are sufficient* to achieve robust TRM training and stable results.
> - *TRM consistently outperforms the majority vote baseline,* though there remains a gap to the upper bound established by pass@$n$.

## 3.2 EXPLOITATION OF TRM

**Setup**    We conduct experiments in two distinct scenarios: (1) answering questions using a search tool, and (2) solving math problems by writing Python code. Following prior works (Jin et al., 2025; Li et al., 2025b), for the search-based QA task, we evaluate on both Qwen2.5-3B-Instruct

---

[7] $\tilde{s}^{n_\tau+1}$ indicates the correctness score for the final answer as predicted by the TRM.

Table 1: Performance of Qwen2.5 variants with different methods on various QA tasks. **Best** results are in bold; second best are underlined.

| Method | General QA | | | | Multi-Hop QA | | | Avg. |
|---|---|---|---|---|---|---|---|---|
| | NQ | TriviaQA | PopQA | HotpotQA | 2wiki | Musique | Bamboogle | |
| *Qwen2.5-3B-Instruct* | | | | | | | | |
| Direct Inference | 12.08 | 32.44 | 13.08 | 15.98 | 24.75 | 2.19 | 2.40 | 14.70 |
| IRCoT | 26.32 | 49.47 | 33.28 | 24.33 | 16.19 | 4.43 | 19.20 | 24.75 |
| RAG | 37.29 | 56.05 | 40.60 | 26.31 | 23.08 | 5.17 | 6.40 | 27.84 |
| SFT | 27.53 | 31.37 | 12.26 | 20.70 | 26.28 | 6.25 | 11.20 | 19.37 |
| R1-PPO | 19.09 | 42.16 | 16.35 | 19.04 | 25.91 | 3.14 | 8.00 | 19.10 |
| R1-GRPO | 25.10 | 45.89 | 18.10 | 21.55 | 27.98 | 5.42 | 20.00 | 23.43 |
| Search-R1-PPO | 36.93 | 54.48 | 35.85 | 32.65 | 32.47 | 12.08 | 24.80 | 32.75 |
| Search-R1-PPO-TRM (ours) | 39.58 | 57.78 | 40.61 | 34.80 | 33.22 | 12.91 | 25.60 | 34.93 |
| Search-R1-GRPO | 47.01 | 61.88 | 45.73 | 43.34 | 42.68 | 18.08 | 37.60 | 42.33 |
| Search-R1-GRPO-TRM (ours) | **47.89** | **62.57** | **47.20** | **44.47** | **43.48** | **19.65** | **39.20** | **43.49** |
| *Qwen2.5-7B-Instruct* | | | | | | | | |
| Direct Inference | 14.29 | 43.69 | 15.10 | 19.23 | 25.54 | 3.68 | 10.40 | 18.85 |
| IRCoT | 18.23 | 50.31 | 30.33 | 21.61 | 8.73 | 4.05 | 17.60 | 21.55 |
| RAG | 34.88 | 58.96 | 39.45 | 30.16 | 23.62 | 5.50 | 21.60 | 30.59 |
| SFT | 31.97 | 34.00 | 12.36 | 22.23 | 26.40 | 9.72 | 10.40 | 21.01 |
| R1-PPO | 22.13 | 49.60 | 17.51 | 22.31 | 28.15 | 6.95 | 30.40 | 25.29 |
| R1-GRPO | 31.61 | 53.69 | 21.60 | 24.96 | 27.47 | 8.77 | 32.00 | 28.59 |
| Search-R1-PPO | 40.86 | 61.42 | 40.15 | 37.84 | 35.27 | 14.81 | 35.20 | 37.94 |
| Search-R1-PPO-TRM (ours) | 43.99 | 61.18 | 41.56 | 39.11 | 37.76 | 17.63 | 38.40 | 39.95 |
| Search-R1-GRPO | 49.97 | 66.81 | 47.59 | 49.06 | **47.80** | 22.30 | **44.80** | 46.90 |
| Search-R1-GRPO-TRM (ours) | **52.11** | **66.90** | **48.52** | **51.32** | 47.67 | **24.99** | **48.80** | **48.62** |

and Qwen2.5-7B-Instruct models; for the code-based math task, we utilize Qwen2.5-Math-1.5B and Qwen2.5-Math-7B (Yang et al., 2024). The training data for each scenario are also sourced from the corresponding prior works to ensure consistency and comparability. The search tool is allowed up to 5 rounds per query, while the code tool can be invoked up to 3 times per problem. All implementations are based on the Verl (Sheng et al., 2025; Zhang et al., 2024) framework. We set $\alpha = 0.05$ for PPO and $\alpha = 0.01$ for GRPO. Additional training details are provided in Appendix B.2.

**Evaluation** For the search scenario, we evaluate performance on both general QA datasets (NQ (Kočiský et al., 2018), TriviaQA (Joshi et al., 2017), PopQA (Mallen et al., 2023)) and multi-hop QA datasets (HotpotQA (Yang et al., 2018), 2Wiki (Ho et al., 2020), Musique (Trivedi et al., 2022), Bamboogle (Press et al., 2023)). For the code-writing scenario, evaluation is conducted on AIME24, AIME25, MATH500 (Hendrycks et al., 2021), Olympiad (He et al., 2024), and AMC23. More evaluation details are in Appendix C.2.

**Baselines** For both search and code scenarios, we consider: (1) `Direct Inference`, which answers questions without any tool usage; (2) `SFT`, supervised fine-tuning without tool usage; and (3) `R1-PPO`/`R1-GRPO`, models trained with PPO or GRPO using outcome-only rewards, without tool usage. Additional baselines for the search scenario include: (1) `RAG`, which retrieves relevant information once before answering; (2) `IRCoT`, iterative retrieval based on previous results; and (3) `Search-R1-PPO`/`Search-R1-GRPO`, trained with PPO or GRPO and allowed to use the search tool. For the code scenario, we further include: (1) `Instruct`, direct inference with the instruct version of Qwen2.5-Math models; (2) `Instruct+PAL` (Gao et al., 2023), generating programs as the intermediate reasoning steps; and (3) `ToRL-PPO`/`ToRL-GRPO`, trained with PPO or GRPO and allowed to use the code tool. More details are shown in Appendix D.

**Results and Analysis** Table 1 and Table 2 summarize the performance of Qwen2.5 variants across different QA and math tasks. Several key observations emerge:

Table 2: Performance of Qwen2.5-Math variants with different methods on various math problems. **Best** results are in bold; second best are underlined.

| Method | AIME24 | AIME25 | MATH500 | Olympiad | AMC23 | Avg. |
|---|---|---|---|---|---|---|
| *Qwen2.5-Math-1.5B* | | | | | | |
| Direct Inference | 7.78 | 1.11 | 67.80 | 28.30 | 35.00 | 28.00 |
| Instruct | 10.67 | 7.22 | 72.60 | 36.59 | **57.50** | 36.92 |
| Instruct+PAL | 34.44 | 0.00 | 21.80 | 10.07 | 17.50 | 16.76 |
| SFT | 0.00 | 0.00 | 15.40 | 7.11 | 27.50 | 10.00 |
| R1-PPO | 11.00 | 10.00 | 74.80 | 33.48 | 55.00 | 36.86 |
| R1-GRPO | 14.11 | 3.67 | 73.40 | 31.70 | **57.50** | 36.08 |
| ToRL-PPO | 19.11 | 13.89 | **75.80** | 43.56 | 55.00 | 41.47 |
| ToRL-PPO-TRM (ours) | **26.00** | 19.89 | **75.80** | 45.78 | 50.00 | 43.49 |
| ToRL-GRPO | 25.56 | 19.33 | **75.80** | 45.19 | 50.00 | 43.18 |
| ToRL-GRPO-TRM (ours) | **26.00** | **27.00** | **75.80** | 45.78 | 52.50 | **45.42** |
| *Qwen2.5-Math-7B* | | | | | | |
| Direct Inference | 12.22 | 6.67 | 69.80 | 30.96 | 40.00 | 31.93 |
| Instruct | 5.11 | 8.11 | 79.60 | 37.33 | 52.50 | 36.53 |
| SFT | 0.00 | 0.00 | 12.80 | 5.19 | 42.50 | 12.10 |
| R1-PPO | 28.11 | 10.11 | 77.40 | 37.93 | 65.00 | 43.71 |
| R1-GRPO | 21.00 | 9.78 | 78.00 | 37.93 | 67.50 | 42.84 |
| ToRL-PPO | 32.56 | 23.11 | 82.60 | **53.04** | 67.50 | 51.76 |
| ToRL-PPO-TRM (ours) | 34.33 | **26.56** | 83.40 | 52.44 | 70.00 | 53.35 |
| ToRL-GRPO | 35.00 | 21.89 | **83.80** | 52.74 | 67.50 | 52.19 |
| ToRL-GRPO-TRM (ours) | **36.56** | 23.67 | 83.20 | 52.59 | **72.50** | **53.70** |

---

**Key Takeaways for TRM Exploitation**

- *TRM consistently enhances model performance in both search and code scenarios, across various model sizes (1.5B, 3B, 7B) and training algorithms (PPO, GRPO)*, indicating that TRM substantially strengthens the ability of LLMs to effectively utilize external tools.

- *Enabling LLMs to dynamically learn tool use yields notable gains*, while reinforcement learning without tool integration leads to much lower performance. Importantly, TRM plays a critical role by helping models utilize tools more effectively.

- *GRPO generally outperforms PPO in our experiments*; however, integrating TRM reliably boosts performance for both approaches.

---

## 3.3 ADDITIONAL ANALYSIS

In this section, we provide further analysis on several key factors related to TRM exploitation and some ablation studies.

**Effect of Hyperparameter $\alpha$**   Figure 5-a shows that for PPO, a very small $\alpha$ limits the effect of TRM, while a very large $\alpha$ overemphasizes tool use. A moderate $\alpha$ balances final performance and reasonable tool invocation. Figure 5-b shows a similar trend for GRPO. We therefore set $\alpha = 0.05$ for PPO and $\alpha = 0.01$ for GRPO in our experiments.

**Improvement of Tool-Use Generalization by TRM**   We investigate the generalization ability of LLMs in tool-use scenarios. Specifically, we evaluate models trained in the search scenario on their ability to use Python code for solving mathematical problems. As shown in Figure 5-c, introducing TRM significantly improves generalization in tool invocation across different scenarios.

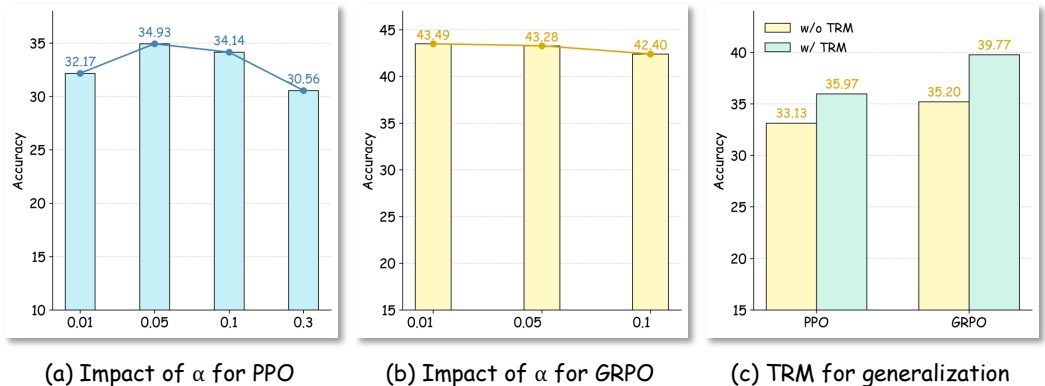

(a) Impact of $\alpha$ for PPO     (b) Impact of $\alpha$ for GRPO     (c) TRM for generalization

Figure 5: Summary of key analysis results. Subfigures (a) and (b) present the influence of the hyperparameter $\alpha$ on PPO and GRPO in conjunction with TRM. Subfigure (c) demonstrates that TRM improves the generalization capability of LLM for tool-use.

**Effect of Turn-Level Advantage Estimation in GRPO**    Unlike turn-level estimation, which normalizes rewards for each turn individually, group-level estimation normalizes all tool-call rewards within a group together (Shao et al., 2024). As shown in Figure 4, turn-level advantage estimation achieves better performance than group-level estimation.

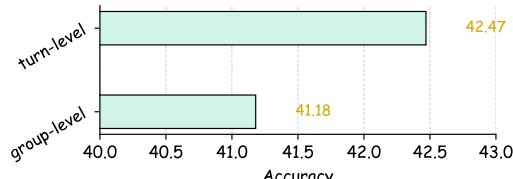

Figure 4: Comparison of group-level and turn-level advantage estimation in GRPO

**Comparison with other process-supervised tool-use methods**    We compare our method with two representative process-supervised baselines: StepSearch (Wang et al., 2025b), which is tailored for search-based QA and evaluates intermediate search queries for relevance and information gain, and AgentPRM (Choudhury, 2025), a general process-supervised method that labels tool calls based on whether they can eventually lead to a correct answer. Figure 6-a shows our method consistently outperforms both baselines in the search scenario over Qwen2.5-3B-Instruct with PPO, highlighting the advantage of our per-tool-call reward modeling over hand-crafted or generic process supervision signals.

**Ablation study to disambiguate distillation and TRM**    To separate the effects of distillation from TRM, we introduce two baselines: ORM, which scores entire trajectories, and TRM used as a verifier, which aggregates per-tool-call scores. Figure 6-b shows that trajectory-level ORM underperforms the answer-only baseline in the search scenario over Qwen2.5-3B-Instruct with PPO, likely because scoring entire trajectories introduces additional noise. TRM as a verifier improves slightly but still lags behind full TRM, suggesting that fine-grained per-tool-call evaluation is essential for guiding the model effectively and fully leveraging the distillation data.

**Ablation study on the necessity and quality of tool calls**    We evaluate the impact of tool-call necessity and quality on model performance and tool usage. Figure 6-c shows that using quality-only yields the lowest performance, likely due to excessive tool usage that introduces noise. Necessity-only reduces the number of tool calls but may compromise the quality of each call, limiting overall effectiveness. Combining both necessity and quality achieves the best performance while maintaining a relatively stable number of tool calls across datasets, suggesting that balancing necessity and quality is important for efficient and effective tool use.

# 4 RELATED WORK

**Process Reward Model**    Reward models have been widely adopted in various reasoning tasks to supervise output quality, such as mathematical problem-solving (Uesato et al., 2023; Shao

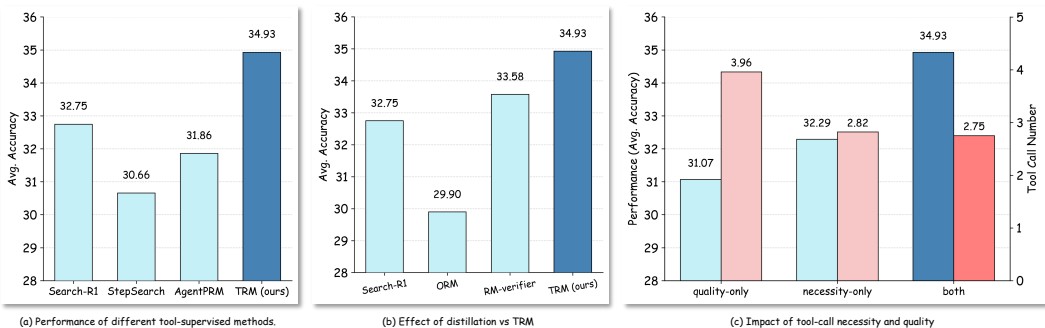

Figure 6: Performance comparisons and ablations for tool-supervised methods.

et al., 2024; Zhang et al., 2025a). These models are generally divided into outcome reward models (ORMs), which provide holistic evaluations, and process reward models (PRMs), which offer fine-grained, step-level assessments. PRMs have shown strong effectiveness (Lightman et al., 2024; Wang et al., 2024; Luo et al., 2024; Cheng et al., 2025), especially in mathematical problem-solving, and have been used both for guiding inference (e.g., best-of-n selection) and for supervising post-training. By providing more granular feedback, process reward models enable models to learn more interpretable and robust reasoning strategies. However, most existing work on PRMs focuses on traditional reasoning tasks, with limited exploration in tool-use scenarios. In this work, we introduce the Tool-call Reward Model (TRM), specifically designed for tool-invocation of LLMs, and conduct a comprehensive study on both the exploration and exploitation of TRM. Our approach aims to extend process-level supervision to agentic tasks, enabling more effective and flexible tool usage in LLMs.

**Agentic RL for LLM Tool Invocation**  Recent advances in outcome-based RL have enabled LLMs to achieve impressive performance in agentic reasoning tasks (Guo et al., 2025; Hu et al., 2025). This paradigm has spurred active research in tool invocation for LLMs, with works such as Search-R1 (Jin et al., 2025), ReSearch (Chen et al., 2025a), R1-Searcher (Song et al., 2025), DeepResearcher (Zheng et al., 2025), WebRL (Qi et al., 2024), WebThinker (Li et al., 2025a), ZeroSearch (Sun et al., 2025), ToRL (Li et al., 2025b), and ToolRL (Qian et al., 2025b) extending outcome-supervised RL to scenarios where LLMs autonomously utilize search engines or code execution for complex reasoning and problem-solving. While these methods have improved agentic capabilities, the reward signals are typically coarse-grained, focusing only on final outcomes and providing limited guidance for efficient tool-use or search strategies. Atom-Searcher (Deng et al., 2025a) and StepSearch (Wang et al., 2025a) further consider intermediate tool-use steps by leveraging existing large models or rule-based approaches. In contrast, our work designs and develops a dedicated TRM to explicitly monitor and supervise intermediate tool invocations, and validates its effectiveness on both search and code-generation scenarios.

## 5  CONCLUSION

We present the Tool-call Reward Model (TRM), a special process reward model that provides fine-grained supervision for tool invocation in large language models. TRM enables more precise credit assignment for each tool call, mitigating issues with outcome-only reward signals such as gradient conflict. We systematically study TRM construction and propose effective integration strategies with classical RL algorithms, including turn-level credit assignment and advantage estimation. Experiments on search-based QA and code-based math tasks show that TRM consistently improves tool usage and generalization across various model sizes and RL methods. Our findings demonstrate that robust TRM performance can be achieved with moderate model sizes and limited training data. We believe TRM offers a promising direction for advancing agentic capabilities in LLMs.

ACKNOWLEDGEMENTS

This work was supported by the China NSFC Projects (62576212, 92370206, U23B2057, 62120106006), and the Shanghai Municipal Science and Technology Project (25X010202846).

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

## THE USE OF LARGE LANGUAGE MODELS

In this work, large language models were used solely for language polishing and improving the clarity of the manuscript. The LLMs did not participate in any substantive aspects of the research, including problem definition, research motivation, methodology, experimental design, or analysis. All scientific contributions, conceptual developments, and experimental results were conducted and validated by the authors without the involvement of LLMs in the core research process.

## ETHICS STATEMENT

This work does not involve sensitive personal data, or practices that raise privacy or security concerns. All datasets used are publicly available and do not contain personally identifiable information. The research does not present potentially harmful methodologies, applications, or insights, and does not raise issues related to discrimination, bias, or fairness. The authors have adhered to the ICLR Code of Ethics throughout the research and submission process.

## REPRODUCIBILITY STATEMENT

All experimental details are provided in the main text (§ 3) and Appendix to ensure reproducibility. Key code components have been submitted with this paper, and the complete codebase will be released publicly at an appropriate time.

## LIMITATIONS AND IMPACTS

While the Tool-call Reward Model (TRM) demonstrates significant improvements in tool-use supervision for large language models, several limitations remain. First, our study is scoped to tasks with verifiable final outcomes (e.g., factual QA and code generation), as our primary focus is to address the limitations of outcome-only reward RL in such settings. Extending TRM to open-ended (Guo et al., 2024) reinforcement learning, where correctness is difficult to assess, would require additional mechanisms for outcome evaluation and is left for future work. Second, to keep rollouts manageable and reduce judge bias, we truncate trajectories to a moderate length, and our current framework does not fully resolve the challenge of providing reliable process supervision for very long tool-use trajectories. Finally, TRM models tool utility via a simple binary necessity–quality decomposition, which may be insufficient to capture more nuanced, multi-objective notions of tool usefulness in complex domains.

Despite these areas for improvement, TRM provides fine-grained supervision that enables more interpretable and robust tool usage, advancing the agentic capabilities of large language models. This approach can facilitate safer and more reliable deployment of LLMs in real-world tasks requiring external tool invocation, and we hope our work inspires further research in process-level reward modeling.

## A PROMPTS

### A.1 PROMPTS FOR TRM TRAINING DATA DISTILLATION

---

**Prompt of Tool Call Evaluation for Search Scenario**

```
## TASK
You are a professional Tool Call Evaluator for AI agent trajectories. For a given
    ↪ user question and its complete step-by-step trajectory, review every tool
    ↪ call (all are of type 'search') and assess each using the following
    ↪evaluation dimensions:
- Tool Selection Accuracy
  - correct (1): It is appropriate to use the 'search' tool for this subtask;
      ↪this call is necessary to make progress.
  - incorrect (0): Using 'search' is not appropriate here (the information is
      ↪already available, the call is unnecessary, or it does not help answer
      ↪the user's question).
- Query Quality
  - perfect (1): The 'query' is clear, directly addresses the user's need, and
      ↪uses precise wording.
  - minor or major error (0):
    - minor error: There is some ambiguity or slight irrelevance, but the search
        ↪will likely still provide useful results.
    - major error: The query is unclear or unrelated to the user's actual need.

## INSTRUCTIONS
- Evaluate every tool call (all are 'search') on both dimensions.
- Briefly justify each score you assign.

## INPUT FORMAT
You will receive:
- 'user_question': The original user question.
- 'trajectory': The full step-by-step trajectory as a list of steps.
  - Each step includes:
    - 'step_id'
    - 'thought': The agent's reasoning or intention before making the search.
    - 'query': The search query issued.
    - 'response': The information returned from the search.
Example:
'''
{{
  "user_question": "What is the capital of France and the population of Germany
      ↪in 2023?",
  "trajectory": [
    {{
      "step_id": 0,
      "thought": "I need to find the capital of France.",
      "query": "capital of France",
      "response": "Paris is the capital of France."
    }},
    {{
      "step_id": 1,
      "thought": "Now I should get the population figure for Germany in 2023.",
      "query": "population of Germany 2023",
      "response": "The population of Germany in 2023 is estimated to be about 84
          ↪million."
    }}
  ]
}}
'''
```

**... (continued in next page)**

---

---

### Prompt of Tool Call Evaluation for Search Scenario (continued)

```
## OUTPUT FORMAT
Provide your evaluations as a JSON list.
For each step, output an object with:
- 'step_id'
- 'tool_selection_accuracy': 1 or 0
- 'tool_selection_justification': your brief justification
- 'query_quality': 1 or 0
- 'query_quality_justification': your brief justification
Example:
'''
[
  {{
    "step_id": 0,
    "tool_selection_accuracy": 1,
    "tool_selection_justification": "The user asked for the capital of France,
        ↪which is factual information requiring a search.",
    "query_quality": 1,
    "query_quality_justification": "The query is clear and directly requests the
        ↪needed information."
  }},
  {{
    "step_id": 1,
    "tool_selection_accuracy": 1,
    "tool_selection_justification": "The user needs the population of Germany in
        ↪2023, which requires a search.",
    "query_quality": 1,
    "query_quality_justification": "The query is specific and unambiguous."
  }}
]
'''

## INPUT
'''
{input}
'''

## OUTPUT
```

---

---

### Prompt of Tool Call Evaluation for Code Scenario

```
### TASK
You are a professional Tool Call Evaluator for AI agent trajectories. For a given
    ↪ user question and its complete step-by-step trajectory, review every tool
    ↪ call (all are of type 'program') and assess each using the following
    ↪evaluation dimensions:
- Tool Selection Accuracy
  - correct (1): It is appropriate to use the 'program' tool for this subtask;
        ↪writing and executing a program is necessary or clearly helpful for
        ↪making progress (e.g., for calculation, verification, or complex
        ↪reasoning).
  - incorrect (0): Using 'program' is not appropriate here (the calculation or
        ↪reasoning can be done easily by hand, the program is unnecessary, or it
        ↪does not help answer the user's question).
- Code Quality
  - perfect (1): The code is complete, correct, and directly serves the intended
        ↪purpose (e.g., correct imports, clear logic, no errors, and directly
        ↪answers the subtask).
  - minor or major error (0):
    - minor error: The progrcodeam has small issues (e.g., missing imports, minor
        ↪ inefficiency), but will likely still work as intended.
    - major error: The code is incomplete, incorrect, or does not address the
        ↪intended purpose.
  **Note**:
  In this evaluation, it is acceptable for the program to be used for verifying
        ↪or checking results that were derived by hand in previous reasoning
        ↪steps. The code does not need to independently derive all intermediate
        ↪parameters or replicate the full logical chain, as long as it correctly
        ↪verifies or computes the intended result. This use of code for auxiliary
        ↪ verification is considered sufficient for a perfect score, provided the
        ↪ code is correct and complete.
### INSTRUCTIONS
- Evaluate every tool call (all are 'program') on both dimensions.
- Briefly justify each score you assign.
### INPUT FORMAT
You will receive:
- 'user_question': The original user question.
- 'trajectory': The full step-by-step trajectory as a list of steps.
  - Each step includes:
    - 'step_id'
    - 'thought': The agent's reasoning or intention before programming.
    - 'code': The code issued (if any; otherwise may be empty).
    - 'output': The output from executing the code (if any; otherwise may be
        ↪empty).
```

**... (continued in next page)**

---

### Prompt of Tool Call Evaluation for Code Scenario (continued)

```
Example:
'''
{{
  "user_question": "What is the sum of the first 100 positive integers?",
  "trajectory": [
    {{
      "step_id": 0,
      "thought": "I can use the formula for the sum of the first n integers, but
            ↪I'll write a program to verify the result.",
      "code": "n = 100\nresult = n * (n + 1) // 2\nprint(result)",
      "output": "5050"
    }},
    {{
      "step_id": 1,
      "thought": "Now I will write a program to sum the integers from 1 to 100
            ↪directly.",
      "code": "print(sum(range(1, 101)))",
      "output": "5050"
    }}
  ]
}}
'''
### OUTPUT FORMAT
Provide your evaluations as a JSON list.
For each step, output an object with:
- 'step_id'
- 'tool_selection_accuracy': 1 or 0
- 'tool_selection_justification': your brief justification
- 'code_quality': 1 or 0
- 'code_quality_justification': your brief justification
Example:
'''
[
  {{
    "step_id": 0,
    "tool_selection_accuracy": 1,
    "tool_selection_justification": "Using a program to verify the formula is
          ↪reasonable and helps ensure correctness.",
    "code_quality": 1,
    "code_quality_justification": "The program is correct, complete, and directly
          ↪ computes the required sum."
  }},
  {{
    "step_id": 1,
    "tool_selection_accuracy": 0,
    "tool_selection_justification": "Writing a second program to do the same
          ↪calculation in a different way is redundant and not necessary for
          ↪solving the user's question.",
    "code_quality": 1,
    "code_quality_justification": "The program is correct and concise, but does
          ↪not add value beyond the previous step."
  }}
]
'''
### INPUT
'''
{input}
'''
### OUTPUT
```

---

### Prompt of Tool Call Evaluation for Multi-tool Scenario

```
## TASK
You are a professional Tool Call Evaluator for AI agent trajectories.
For a given user question, a list of available tools (with descriptions and
    ↪parameter schemas),
and the complete step-by-step trajectory, review every tool call and assess each
    ↪using the
following evaluation dimensions:
- Tool Selection Accuracy
  - correct (1):
    - The chosen tool matches the intended subtask and is consistent with the
        ↪tool's description and schema.
    - The call is necessary or clearly helpful for making progress toward
        ↪answering the user's question or fulfilling the user's request.
  - incorrect (0):
    - The chosen tool does **not** match the subtask (e.g., wrong tool given the
        ↪intention or user need).
    - Or the call is redundant / unnecessary (e.g., the information is already
        ↪available from earlier steps, or the call does not help answer the user
        ↪'s question).
- Argument Quality
  - perfect (1):
    - The arguments to the tool are correct, complete, and specific.
    - They respect the tool's parameter schema (types, required fields) and align
        ↪ with the user's need or the agent's stated intention.
  - minor or major error (0):
    - minor error:
      - Small mismatch, ambiguity, or slight irrelevance in arguments that still
          ↪likely allows the tool to work and return useful results.
    - major error:
      - Missing required fields, wrong types, wrong values, or arguments that do
          ↪not actually reflect the intended subtask or the user request.
      - The tool would likely fail, error, or return irrelevant / unusable
          ↪results.

## INSTRUCTIONS
- Evaluate every tool call on both dimensions.
- Briefly justify each score you assign.

## INPUT FORMAT
You will receive:
- 'user_question': The original user question.
- 'trajectory': The full step-by-step trajectory as a list of steps.
  - Each step includes:
    - 'step_id'
    - 'thought': The agent's reasoning or intention before making the search.
    - 'tool_calls': The tools invoked.
    - 'response': The information returned from the tool calls.
- 'available_tools': A list of available tools with their descriptions and
    ↪parameter schemas.
```

**... (continued in next page)**

---

**Prompt of Tool Call Evaluation for Multi-tool Scenario (continued)**

```
Example:
```
{{
  "user_question": "What is the capital of France and the population of Germany
      ↪in 2023?",
  "available_tools": [
    {{
      "name": "search",
      "description": "Use this tool to search for factual information from the
          ↪web.",
      "parameters": {{
        "type": "object",
        "properties": {{
          "query": {{
            "type": "string",
            "description": "The search query string."
          }}
        }},
        "required": ["query"]
      }}
    }}
  ],
  "trajectory": [
    {{
      "step_id": 0,
      "thought": "I need to find the capital of France.",
      "tool_calls": [{{
        "name": "search",
        "arguments": {{
          "query": "capital of France"
        }}
      }}],
      "response": "Paris is the capital of France."
    }},
    {{
      "step_id": 1,
      "thought": "Now I should get the population figure for Germany in 2023.",
      "tool_calls": [{{
        "name": "search",
        "arguments": {{
          "query": "population of Germany 2023"
        }}
      }}],
      "response": "The population of Germany in 2023 is estimated to be about 84
          ↪million."
    }}
  ]
}}
```

**... (continued in next page)**

---

**Prompt of Tool Call Evaluation for Multi-tool Scenario (continued)**

```
## OUTPUT FORMAT
Provide your evaluations as a JSON list.
For each step, output an object with:
- 'step_id'
- 'tool_selection_accuracy': 1 or 0
- 'tool_selection_justification': your brief justification
- 'argument_quality': 1 or 0
- 'argument_quality_justification': your brief justification
Example:
'''
[
  {{
    "step_id": 0,
    "tool_selection_accuracy": 1,
    "tool_selection_justification": "The user asked for the capital of France,
        ↪which is factual information requiring a search.",
    "argument_quality": 1,
    "argument_quality_justification": "The query is clear and directly requests
        ↪the needed information."
  }},
  {{
    "step_id": 1,
    "tool_selection_accuracy": 1,
    "tool_selection_justification": "The user needs the population of Germany in
        ↪2023, which requires a search.",
    "argument_quality": 1,
    "argument_quality_justification": "The query is specific and unambiguous."
  }}
]
'''

## INPUT
'''
{input}
'''

## OUTPUT
```

---

### A.2  SYSTEM PROMPTS FOR TASKS

---

**System Prompt for QA Tasks with Search Tool**

```
Answer the given question. You must conduct reasoning inside <think> and </think>
    ↪ first every time you get new information. After reasoning, if you find
    ↪you lack some knowledge, you can call a search engine by <search> query </
    ↪search> and it will return the top searched results between <tool_response
    ↪> and </tool_response>. You can search as many times as your want. If you
    ↪find no further external knowledge needed, you can directly provide the
    ↪answer inside <answer> and </answer>, without detailed illustrations. For
    ↪example, <answer> Beijing </answer>.
```

### System Prompt for QA Tasks without Search Tool

```
Answer the given question. You should first have a reasoning process in mind and
    ↪then provides the answer. Show your reasoning in <think> </think> tags and
    ↪ return the final answer in <answer> </answer> tags, for example <answer>
    ↪Beijing </answer>.
```

### System Prompt for Mathematical Problems with Code Tool

```
Solve the following problem step by step. You now have the ability to selectively
    ↪ write executable Python code to enhance your reasoning process. The
    ↪Python code will be executed by an external sandbox, and the output (
    ↪wrapped in '<tool_response>output_str</tool_response>') can be returned to
    ↪ aid your reasoning and help you arrive at the final answer. The Python
    ↪code should be complete scripts, including necessary imports. Put your
    ↪final answer within \\boxed{}.
```

### System Prompt for Mathematical Problems without Code Tool

```
Please reason step by step, and put your final answer within \\boxed{}.
```

### System Prompt for Real-world Problems with Multi Tools

```
In this environment you have access to a set of tools you can use to assist with
    ↪the user query. You may perform multiple rounds of function calls. In each
    ↪ round, you can call one or more functions.

Here are available functions in JSONSchema format:
'''json
{func_schemas}
'''

In your response, you need to first think about the reasoning process in the mind
    ↪ and then conduct function calling to get the information or perform the
    ↪actions if needed. The reasoning process and function calling are enclosed
    ↪ within <think> </think> and <tool_call> </tool_call> tags. The results of
    ↪ the function calls will be given back to you after execution, and you can
    ↪ continue to call functions until you can provide the final answer
    ↪enclosed within <answer> </answer> tags for the user's question.

Tool call example:
<tool_call>
{{"name": <function-name>, "arguments": <args-json-object>}}
...
</tool_call>

Final answer example:
<answer> ... </answer>
```

## B   TRAINING DETAILS

### B.1   TRM TRAINING DETAILS

For the search scenario, we use Qwen2.5-3B-Instruct as the backbone model and train with 10K examples. For the code scenario, we adopt Qwen2.5-Math-1.5B as the backbone and utilize 20K training samples. The training process follows standard supervised fine-tuning procedures, and the key hyperparameters are summarized as follows. We set the number of training epochs to 10, with a learning rate of 1e-6. The global batch size 128. The maximum sequence length is 8192, and the maximum prompt length is 1024. All experiments are conducted using Huggingface implementation[8].

### B.2   RL DETAILS

**Tool Execution Environment**   For the search tool, we follow the setup of Search-R1 (Jin et al., 2025) and use the 2018 Wikipedia dump (Karpukhin et al., 2020) as the knowledge source. The E5 (Wang et al., 2022) retriever is employed to retrieve relevant passages for each query. For the Python code execution environment, we follow the approach in ToRL (Li et al., 2025b) and utilize the SandboxFusion environment (Bytedance-Seed-Foundation-Code-Team et al., 2025) to safely execute code snippets. This setup ensures both the reliability and security of tool interactions during reinforcement learning experiments.

Table 3: Hyperparameters in RL. The notation *3B / 1.5B* and *7B / 7B* denote the backbone model sizes used for different tasks: the first value corresponds to the search tool for QA, and the second value corresponds to the Python code tool for mathematical problem solving.

| Hyperparameter | PPO | | GRPO | |
|---|---|---|---|---|
| | *3B / 1.5B* | *7B / 7B* | *3B / 1.5B* | *7B / 7B* |
| `trainer.total_training_steps` | 500 | 300 | 500 | 300 |
| `algorithm.adv_estimator` | gae | gae | grpo | grpo |
| `data.train_batch_size` | 512 | 512 | 512 / 128 | 512 / 128 |
| `actor_rollout_ref.actor.ppo_mini_batch_size` | 256 | 256 | 256 / 64 | 256 / 64 |
| `data.max_prompt_length` | 8192 / 3072 | 8192 / 3072 | 8192 / 3072 | 8192 / 3072 |
| `data.max_response_length` | 512 / 1024 | 512 / 1024 | 512 / 1024 | 512 / 1024 |
| `tools.max_tool_resp_len` | 512 | 512 | 512 | 512 |
| `actor_rollout_ref.actor.optim.lr` | 2e-7 / 1e-6 | 2e-7 / 1e-6 | 2e-6 | 2e-6 |
| `critic.optim.lr` | 5e-7 / 5e-6 | 5e-7 / 5e-6 | - | - |
| `actor_rollout_ref.actor.entropy_coeff` | 0.001 | 0.001 | 0 | 0 |
| `actor_rollout_ref.rollout.temperature` | 1.0 | 1.0 | 1.0 | 1.0 |
| `actor_rollout_ref.rollout.n` | 1 | 1 | 5 / 8 | 5 / 8 |
| `tools.max_turns` | 5 / 3 | 5 / 3 | 5 / 3 | 5 / 3 |
| `algorithm.kl_ctrl.kl_coef` | 0.001 | 0.001 | - | - |
| `actor_rollout_ref.actor.kl_loss_coef` | - | - | 0.001 | 0.001 |

**Training Hyperparameters**   Table 3 presents the key hyperparameters used in our RL experiments. All other training configurations follow standard practices as described in the main text.

## C   EVALUATION DETAILS

### C.1   TRM EVALUATION

For evaluation, we use the checkpoint[9] from Search-R1 (Jin et al., 2025) to collect rollout candidates from the prompts in validation sets of HotpotQA and 2wikimultihopQA. During rollout generation, we set the sampling temperature to 0. Additionally, the agent is allowed to perform up to 3 search steps per query.

---

[8] https://huggingface.co/docs/trl/prm_trainer
[9] https://huggingface.co/PeterJinGo/SearchR1-nq_hotpotqa_train-qwen2.5-7b-em-ppo-v0.2

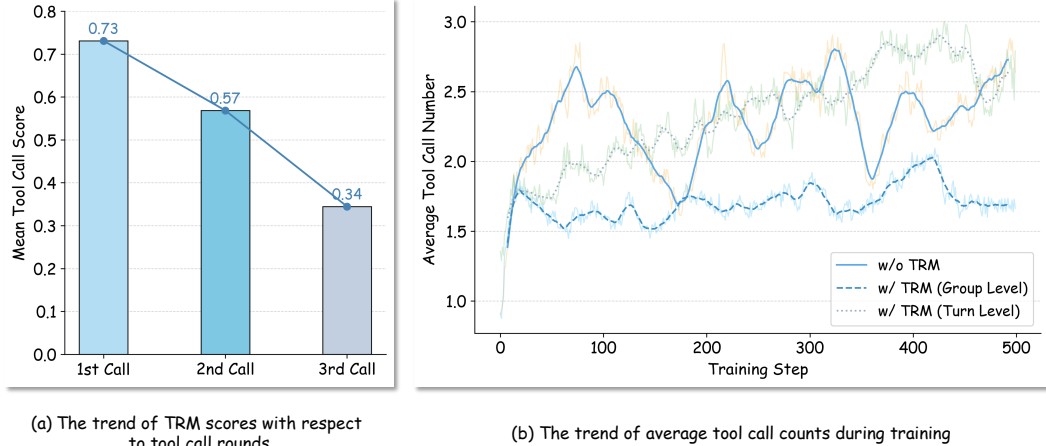

(a) The trend of TRM scores with respect to tool call rounds

(b) The trend of average tool call counts during training

Figure 7: Reward hacking in group-level advantage estimation for GRPO: (a) TRM scores decrease with more tool calls, and (b) turn-level estimation mitigates the penalization of longer tool call sequences.

## C.2 LLM Evaluation

For LLM evaluation, we set the sampling temperature to 0 to encourage deterministic generation. [10] Other parameters, such as the maximum number of tool calls, are kept consistent with those used during training. Notably, since AIME24 and AIME25 contain very few problems, we report the average results over 30 repeated evaluations for these two datasets to ensure statistical reliability.

## D    Baseline Details

**Training-free Methods**   For IRCoT and RAG, we mainly use the implementation[11] of Re-Search (Chen et al., 2025a).

**Training Methods**   For SFT, we adopt LLaMA-Factory (Zheng et al., 2024). For R1, we simply disable tool invocation in our framework.

## E    Additional Experimental Results

### E.1    Reward Hacking Caused by Group-level Advantage Estimation

Group-level advantage estimation in GRPO can lead to reward hacking, where the model prefers shorter tool call sequences. This is because cascading errors make later tool calls less reliable, resulting in lower scores and penalization for longer sequences (Figure 7-a). In contrast, turn-level advantage estimation alleviates this issue by treating each tool call independently, encouraging more stable tool usage (Figure 7-b). Tool-call numbers on evaluation benchmarks in Table 4 are consistent. Notably, introducing TRM does not significantly increase the number of tool calls compared to outcome-only training methods.

### E.2    Resource Overhead Introduced by TRM

As shown in Table 5, incorporating TRM introduces only an $8.8\%$ overhead per training step, indicating minimal additional compute cost. BoN inference experiences a $50\%$ increase in per-sample time with TRM; however, the absolute time remains small (with the full BoN evaluation taking $\sim 20$ minutes), which is practically negligible.

---

[10]Due to the use of the vLLM server, some randomness may still be present during evaluation.

[11]https://github.com/bytedance/SandboxFusion/tree/main

Table 4: Average tol-call numbers on various QA tasks over Qwen2.5-3B-Instruct with PPO: turn-level vs. group-level

| Method | General QA | | | Multi-Hop QA | | | | Avg. |
|--------|-----|----------|-------|----------|-------|---------|-----------|------|
| | NQ | TriviaQA | PopQA | HotpotQA | 2wiki | Musique | Bamboogle | |
| Group-level | 1.99 | 1.93 | 2.00 | 2.42 | 2.81 | 3.01 | 2.37 | 2.36 |
| Turn-level | 2.59 | 2.44 | 2.48 | 2.68 | 3.01 | 3.19 | 2.68 | 2.72 |

Table 5: Training and BoN inference speed with vs. without TRM on Qwen2.5-3B-Instruct with 8xA800 GPUs under PPO

| Method | Training (s/step) | BoN Inference (s/sample) |
|--------|-------------------|--------------------------|
| w/o TRM | 56.9 | 0.14 |
| w/ TRM | 61.9 (+8.8%) | 0.21 (+50%) |

### E.3 TRM Training Data Quality Verification

Regarding TRM training data quality, we randomly sampled 100 trajectories and evaluated them using both human annotators and a multi-model judge (DeepSeek-R1, Qwen2.5-72B-Instruct, and GPT-4o-mini). As shown in Table 6, the resulting dataset demonstrates relatively high quality.

Table 6: TRM Training Data Quality Evaluation (Accuracy)

| Evaluation Method | Necessity (%) | Quality (%) |
|-------------------|---------------|-------------|
| Human | 89 | 89 |
| Multi-model | 86 | 75 |

### E.4 Generality Beyond Search and Code Tools

We follow the ReCall (Chen et al., 2025a) work to evaluate our method in more diverse multi-tool scenarios. These scenarios include tasks such as library management, employee management, and travel planning, which require the use of multiple APIs (e.g., searching and reserving books, calculating salaries, updating flight bookings, etc.). These additional experiments (see Table 7) show that introducing TRM continues to provide consistent improvements in broader multi-tool settings.

Table 7: Performance in more diverse multi-tool scenarios on Qwen2.5-7B-Instruct with GRPO

| Method | Qwen2.5-7B-Instruct | ReCall | ReCall-TRM |
|--------|---------------------|--------|------------|
| F1 | 7.37 | 39.71 | 43.28 |

### E.5 Case Study

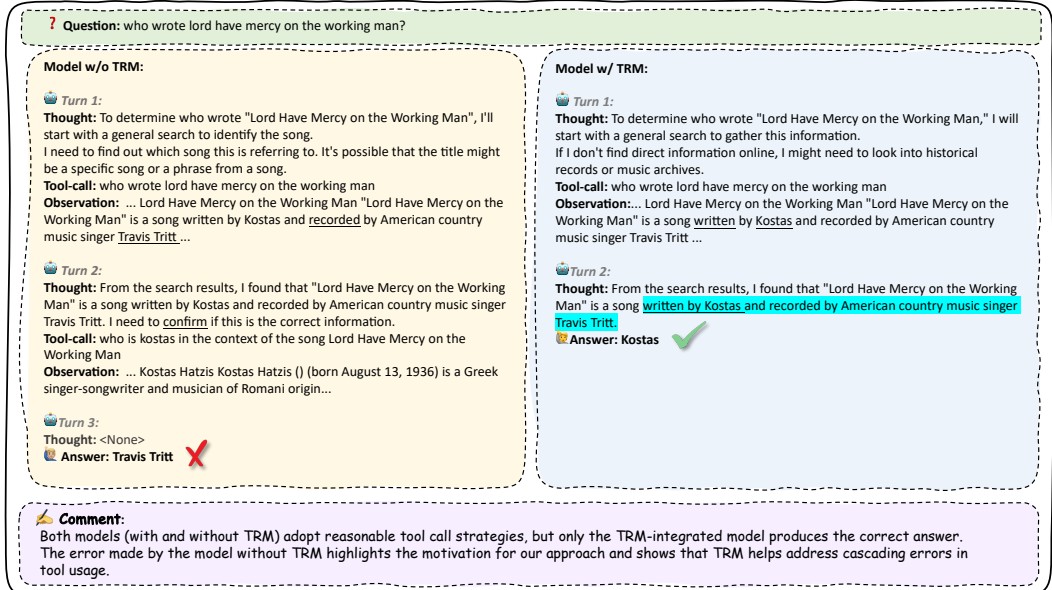

Figure 8: Case 1

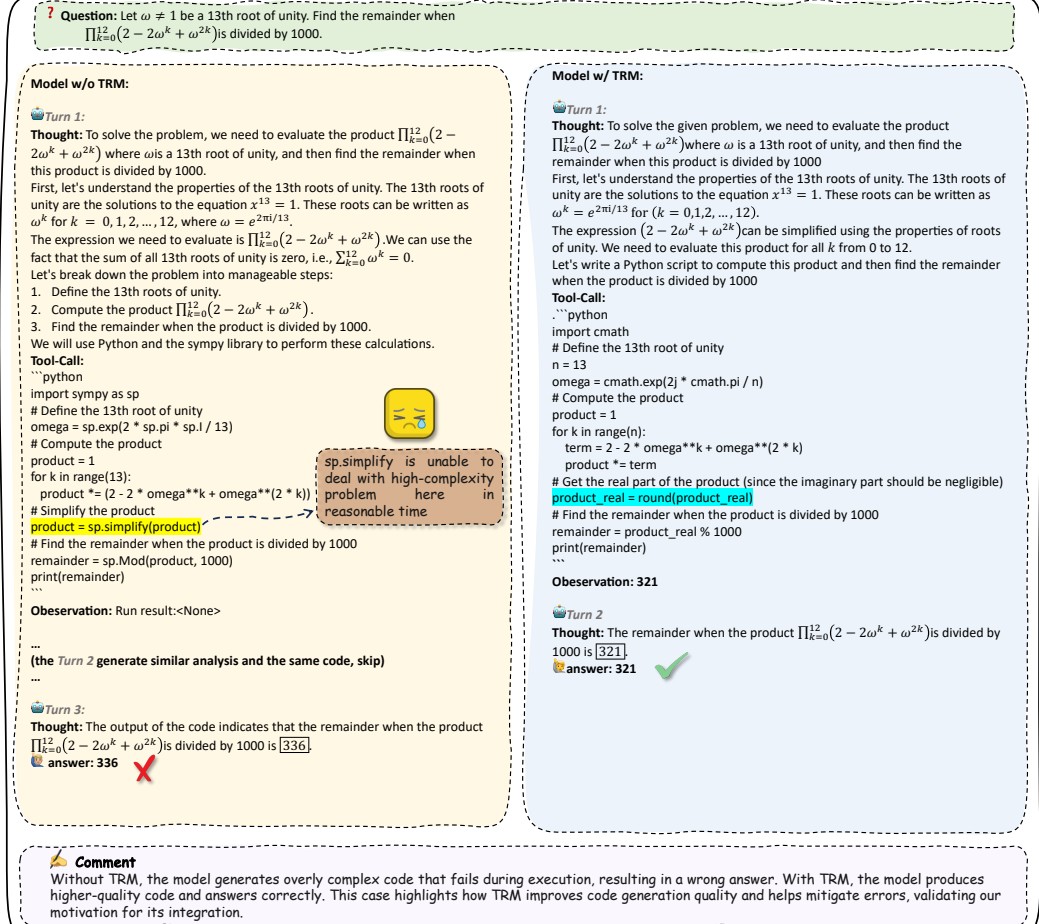

Figure 9: Case 2

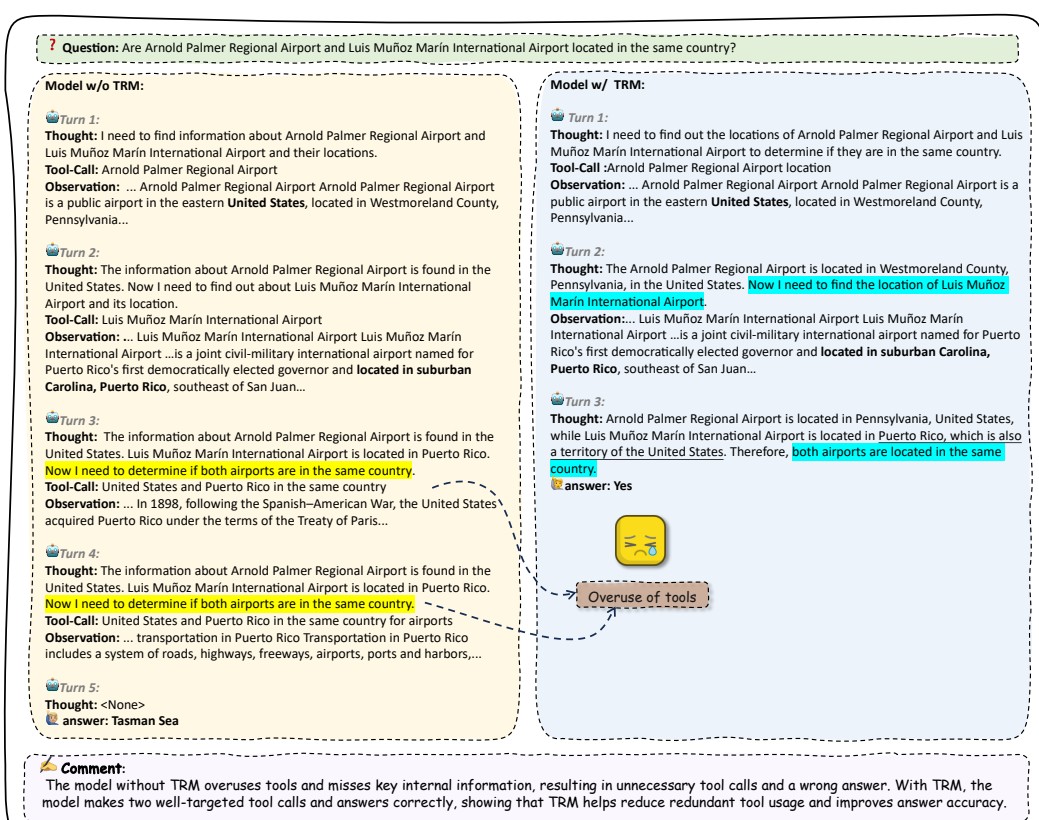

Figure 10: Case 3

## E.6 PROCESS-LEVEL TRANSPARENCY OF TRM

Table 8: Examples of failed tool calls

| Question | Tool Calls | Comment | Explanation |
|---|---|---|---|
| What was Bill Dvorak's fifth cousin's nationality? | **search**(Bill Dvorak nationality) | low-quality | The query asks about Bill Dvorak's nationality which is unrelated to the user's need (fifth cousin's nationality). It reflects a major misinterpretation of the question. |
| Are Arnold Palmer Regional Airport and Luis Muñoz Marín International Airport located in the same country? | **search**(Arnold Palmer Regional Airport) **search**(United States and Puerto Rico in the same country) **search**(United States and Puerto Rico in the same country for airports) | unnecessary | Invoke duplicate tools |
| The product $8 * .25 * 2 * .125 =$ | # Calculate the product step by step

result = 8 * 0.25 * 2 * 0.125

print(result) | unnecessary | The multiplication of four numbers (8, 0.25, 2, 0.125) is simple and can be easily computed by hand; writing a program is unnecessary for this straightforward arithmetic. |
| Medians $BD$ and $CE$ of triangle $ABC$ are perpendicular, $BD=8$, and $CE=12$. The area of triangle $ABC$ is | Very long code ... | low-quality | The code contains significant unused symbolic logic (Sympy setup/solving) that's irrelevant to the final arithmetic operation, which is inefficient and shows flawed implementation. Though the output is correct, the dead code constitutes a major structural issue. |

## E.7 DETAILED EXPERIMENTAL RESULTS

Table 9: Detailed results of Figure 3

| Method | Model Size | Training Data Size | 1 | 2 | 3 | 4 | 5 | Avg |
|---|---|---|---|---|---|---|---|---|
| *2wikimultihopqa [out-of-domain] (dev split)* | | | | | | | | |
| Majority Vote | – | – | 11.73 | 11.73 | 10.82 | 10.76 | 10.56 | 11.12 |
| Pass@n | | | 11.73 | 15.16 | 17.40 | 19.02 | 20.26 | 16.71 |
| TRM | 3B | 1K | 11.73 | 13.53 | 14.58 | 15.33 | 15.76 | 14.19 |
| | | 2K | 11.73 | 13.79 | 15.02 | 15.80 | 16.39 | 14.55 |
| | | 5K | 11.73 | 14.33 | 15.94 | 16.94 | 17.80 | 15.35 |
| | | 10K | 11.73 | 14.29 | 15.92 | 16.94 | 17.66 | 15.31 |
| TRM | 0.5B | 10K | 11.73 | 12.31 | 12.81 | 13.10 | 13.20 | 12.63 |
| | 1.5B | | 11.73 | 13.88 | 15.17 | 15.97 | 16.68 | 14.69 |
| | 7B | | 11.73 | 13.36 | 14.54 | 15.20 | 15.66 | 14.10 |
| *hotpotqa [in-domain] (dev split)* | | | | | | | | |
| Majority Vote | – | – | 24.52 | 24.52 | 25.05 | 25.08 | 25.29 | 24.89 |
| Pass@n | | | 24.52 | 30.97 | 33.96 | 35.83 | 37.54 | 32.56 |
| TRM | 3B | 1K | 24.52 | 28.16 | 29.74 | 30.40 | 31.02 | 28.77 |
| | | 2K | 24.52 | 26.85 | 27.93 | 28.66 | 28.82 | 27.36 |
| | | 5K | 24.52 | 28.82 | 30.70 | 31.61 | 32.40 | 29.61 |
| | | 10K | 24.52 | 29.03 | 30.90 | 32.14 | 32.88 | 29.89 |
| TRM | 0.5B | 10K | 24.52 | 25.23 | 25.28 | 25.21 | 24.85 | 25.02 |
| | 1.5B | | 24.52 | 27.86 | 29.40 | 30.26 | 30.71 | 28.55 |
| | 7B | | 24.52 | 26.91 | 27.75 | 28.53 | 28.94 | 27.33 |

Table 10: Detailed results of Figure 5-a and Figure 5-b

| Method | $\alpha$ | NQ | TriviaQA | PopQA | HotpotQA | 2wiki | Musique | Bamboogle | Avg. |
|---|---|---|---|---|---|---|---|---|---|
| PPO | 0.01 | 38.14 | 55.20 | 36.17 | 32.64 | 31.00 | 11.21 | 20.80 | 32.17 |
| | 0.05 | 39.58 | 57.78 | 40.61 | 34.80 | 33.22 | 12.91 | 25.60 | 34.93 |
| | 0.1 | 40.08 | 55.82 | 39.11 | 32.91 | 32.73 | 11.12 | 27.20 | 34.14 |
| | 0.3 | 34.52 | 50.08 | 35.98 | 29.41 | 26.81 | 9.10 | 28.00 | 30.56 |
| GRPO | 0.01 | 47.89 | 62.57 | 47.20 | 44.47 | 43.48 | 19.65 | 39.20 | 43.49 |
| | 0.05 | 48.09 | 63.04 | 46.93 | 44.66 | 43.45 | 19.20 | 37.60 | 43.28 |
| | 0.1 | 46.68 | 62.58 | 45.93 | 43.47 | 42.89 | 16.88 | 38.40 | 42.40 |

Table 11: Detailed results of Figure 5-c

| Method | Search-2 Code | | | Avg. |
|---|---|---|---|---|
| | MATH500 | Olympiad | AMC23 | |
| ToRL-PPO | 50.40 | 24.00 | 25.00 | 33.13 |
| ToRL-PPO-TRM (ours) | 54.20 | 26.22 | 27.50 | 35.97 |
| ToRL-GRPO | 52.80 | 22.81 | 30.00 | 35.20 |
| ToRL-GRPO-TRM (ours) | 56.60 | 27.70 | 35.00 | 39.77 |

Table 12: Detailed results of Figure 6

| Method | NQ | TriviaQA | PopQA | HotpotQA | 2wiki | Musique | Bamboogle | Avg. |
|---|---|---|---|---|---|---|---|---|
| *Performance* | | | | | | | | |
| Search-R1 + StepSearch | 37.53 | 55.17 | 39.20 | 29.75 | 27.65 | 7.74 | 17.60 | 30.66 |
| Search-R1 + AgentPRM | 38.01 | 54.62 | 37.07 | 32.78 | 31.15 | 10.22 | 19.20 | 31.86 |
| Search-R1 + ORM | 39.47 | 56.17 | 40.93 | 29.63 | 26.65 | 6.83 | 9.60 | 29.90 |
| Search-R1 + TRM-verifier | 35.65 | 54.10 | 36.46 | 33.91 | 33.90 | 13.86 | 27.20 | 33.58 |
| quality-only | 38.95 | 56.17 | 38.94 | 31.06 | 27.93 | 8.44 | 16.00 | 31.07 |
| necessity-only | 37.42 | 54.06 | 37.21 | 32.46 | 31.80 | 11.46 | 21.60 | 32.29 |
| *Tool-call Number* | | | | | | | | |
| quality-only | 3.96 | 3.94 | 3.93 | 3.96 | 3.98 | 3.99 | 3.93 | 3.96 |
| necessity-only | 2.58 | 2.68 | 2.58 | 2.83 | 2.95 | 3.31 | 2.84 | 2.82 |
| both | 2.37 | 2.39 | 2.37 | 2.85 | 3.35 | 3.23 | 2.71 | 2.75 |

