# OpenReview forum: "Empowering LLM Tool Invocation with Tool-call Reward Model"
_ICLR.cc/2026/Conference — ICLR 2026 Poster_

### Official Review · Reviewer_R2Sf · 2025-10-31

**Soundness:** 3
**Presentation:** 3
**Contribution:** 3
**Rating:** 4
**Confidence:** 4

**Summary:**

The paper introduces TRM, a Tool-call Reward Model for LLMs tool learning. TRM is a process reward model that assigns per-tool-call rewards based on necessity and quality, so that it overcomes the limitations of outcome reward signals that provide credit only for the final result. The paper describes methods for constructing effective TRM model and introduces integration strategies with RL algorithms including PPO and GRPO using refined credit assignment and turn-level advantage estimation. Experiments on both search-based QA and code-based math tasks demonstrate robust gains over RL with outcome rewards.

**Strengths:**

- The problem of designing a PRM for tool-integrated reasoning is well motivated and very timely.
- The reward function designed to evaluate necessity and quality is reasonable, and empirical results showed it worked well.
- Empirical results showed that the proposed reward functions worked pretty well on search and code tool related datasets.

**Weaknesses:**

- The eval on search tool and code tool are relatively thorough. Although these are important, the approach is advertised as broadly applicable—yet evidence supporting cross-task generalization is mainly anecdotal, and the set of tools remains narrow. A more ambitious empirical scope (e.g., more diverse tool types, additional real-world external APIs) would strengthen the claim of general utility.
- TRM relies on a strong model (DeepSeek-R1) to annotate and produce the actual reward values for both necessity and quality. This essentially brings extra knowledge and inductive bias into the training process, which might be a bit unfair. The compared methods (including Search-R1) only use the training dataset (the ground truth answer label).
- The baseline missed some leading program-aided tool-integration works (e.g., PAL, StableToolBench).

Minor issue:
1. To be more precise, a reward function instead of a reward model is proposed in this paper to guide more effective/efficient tool calls. There is not really a reward model being trained to guide something that is unverifiable in the answer and needs tool call. As a result, I do not think the "Reward Model" in the paper title is appropriate.

**Questions:**

- In principle, the reward function designed in the paper encourages the base model to act both effectively (quality) and efficiently (necessity). Most of the evals in the paper are about effectiveness regarding the answer accuracy. Is there any eval regaring the efficiency of tool use by the learned model? For example, the learned model should use less tool calls to get the correct answer compared to baseline methods.
- How would the proposed TRM construction pipeline adapt to tool types with more complex or less binary notions of utility?
- Have you analyzed failure cases of TRM or inspected which features/representations contribute most to per-tool-call utility predictions, possibly to increase process-level transparency?

---

> ### Author Response · Authors · 2025-11-23
> **Response to Reviewer R2Sf (part 1/3)**
>
> Thank you very much for the time and effort you devoted to our work. Your insightful comments have greatly strengthened our work.
>
> ---
>
> ### On the Generality Beyond Search and Code Tools ---> Weakness 1
> We agree that it is important to provide evidence beyond search and code tools in order to support the claim of broader applicability. In addition to the search and code settings, we also follow ReCall [1] and evaluate our method in their multi-tool scenarios, which include tasks such as library management, employee management, and travel planning. These tasks require coordinating multiple APIs (e.g., searching and reserving books, computing salaries, updating or cancelling flight bookings, etc.), and therefore go beyond a narrow, single-tool setting.
>
> As shown in Table 1, adding TRM on top of ReCall leads to consistent gains in these more diverse multi-tool scenarios. This provides more systematic evidence that our approach can generalize across different tasks and tool combinations, rather than being limited to search and code tools only.
>
> Relevant results have been updated in lines 1432–1444 of the revised manuscript.
>
> **Table 1: Performance in More Diverse Multi-Tool Scenarios on Qwen2.5-7B-Instruct**
> | Method                       | F1    |
> | ---------------------------- | ----- |
> | Qwen2.5-7B-Instruct          | 7.37  |
> | ReCall (outcome-reward-only) | 39.71 |
> | ReCall + TRM                 | 43.28 |
>
> ---
>
> ### Ablation Study to Disambiguate Distillation and TRM ---> Weakness 2
> We greatly appreciate this constructive suggestion. To disambiguate the effects of distillation and TRM, we introduce two additional baselines:
> * *Outcome Reward Model (ORM)*: We train an ORM on the same data used to train TRM. ORM provides a single score for each entire trajectory, which is then combined with answer correctness via a weighted sum.
> * *TRM as a verifier*: We use the trained TRM to score trajectories, where the geometric mean of all per-tool-call scores along a trajectory is combined with answer correctness.
>
> Both baselines leverage the distillation data; the key distinction is that TRM provides fine-grained, per-tool-call evaluation signals, allowing us to clearly separate the contributions of distillation from those of TRM.
>
> Table 2 highlights the effects of distillation and TRM. Using a trajectory-level ORM (Search-R1-ORM) underperforms even the answer-only baseline (Search-R1 [2]), likely because ORM introduces additional noise from answer evaluation. Using TRM as a verifier (Search-R1-TRM-verifier) yields some improvement over answer-only evaluation, but still falls short of the full TRM model. These results demonstrate that the fine-grained per-tool-call signals provided by TRM are crucial for its effectiveness, clearly distinguishing its contribution from that of distillation.
>
> Relevant results have been updated in lines 471–478 of the revised manuscript.
>
> **Table 2: Ablation Study to Disambiguate Distillation and TRM on Qwen2.5-3B**
> | Method                 | NQ    | TriviaQA | PopQA | HotpotQA | 2wiki | Musique | Bamboogle | Avg.  |
> | ---------------------- | ----- | -------- | ----- | -------- | ----- | ------- | --------- | ----- |
> | Search-R1              | 36.93 | 54.48    | 35.85 | 32.65    | 32.47 | 12.08   | 24.80     | 32.75 |
> | Search-R1-ORM          | 39.47 | 56.17    | 40.93 | 29.63    | 26.65 | 6.83    | 9.60      | 29.90 |
> | Search-R1-TRM-verifier | 35.65 | 54.10    | 36.46 | 33.91    | 33.90 | 13.86   | 27.20     | 33.58 |
> | Search-R1-TRM (ours)   | 39.58 | 57.78    | 40.61 | 34.80    | 33.22 | 12.91   | 25.60     | 34.93 |
>
> [1] https://attractive-almandine-935.notion.site/ReCall-Learning-to-Reason-with-Tool-Call-for-LLMs-via-Reinforcement-Learning-1d7aec91e9bb8006ad40f9edbfe2191a
>
> [2] Bowen Jin, Hansi Zeng, Zhenrui Yue, Jinsung Yoon, Sercan O Arik, Dong Wang, Hamed Zamani, and Jiawei Han. Search-r1: Training LLMs to reason and leverage search engines with reinforcement learning. In Second Conference on Language Modeling, 2025.

---

> ### Author Response · Authors · 2025-11-23
> **Response to Reviewer R2Sf (part 2/3)**
>
> ### On Evaluating the Efficiency of Tool Use ---> Question 1
> Thank you for pointing this out. We agree that, beyond answer accuracy, it is important to evaluate whether the learned policy model uses tools efficiently. In addition to the accuracy metrics, we conduct an ablation study that explicitly disentangles the effects of the quality and necessity components in our reward design. Table 3 reports answer accuracy under three variants: (i) quality-only, (ii) necessity-only, and (iii) both (our full method). Table 4 reports the average number of tool calls per example under the same settings.
>
> The results show that:
> * Quality-only achieves the lowest accuracy and also overuses tools (e.g., an average of 3.96 calls vs. 2.75 for the full method), indicating that optimizing only for answer quality encourages unnecessary tool usage.
> * Necessity-only indeed reduces the number of tool calls (2.82 on average), but this comes at a cost in accuracy, suggesting that being too “stingy” with tools harms the quality of reasoning.
> * Combining both necessity and quality yields the best overall accuracy (Avg. 34.93 vs. 31.07/32.29) while keeping the average number of tool calls relatively low and stable across datasets (2.75 calls on average, close to the necessity-only variant and much lower than quality-only).
>
> Relevant results have been updated in lines 480–485 of the revised manuscript.
>
> **Table 3: Model Performance Ablation on Tool-Call Necessity and Quality (Qwen2.5-3B-Instruct)**
> | Method         | NQ    | TriviaQA | PopQA | HotpotQA | 2wiki | Musique | Bamboogle | Avg.  |
> | -------------- | ----- | -------- | ----- | -------- | ----- | ------- | --------- | ----- |
> | quality-only   | 38.95 | 56.17    | 38.94 | 31.06    | 27.93 | 8.44    | 16.00     | 31.07 |
> | necessity-only | 37.42 | 54.06    | 37.21 | 32.46    | 31.80 | 11.46   | 21.60     | 32.29 |
> | both           | 39.58 | 57.78    | 40.61 | 34.80    | 33.22 | 12.91   | 25.60     | 34.93 |
>
> **Table 4: Tool-Call Number Ablation on Necessity and Quality (Qwen2.5-3B-Instruct)**
> | Method         | NQ   | TriviaQA | PopQA | HotpotQA | 2wiki | Musique | Bamboogle | Avg. |
> | -------------- | ---- | -------- | ----- | -------- | ----- | ------- | --------- | ---- |
> | quality-only   | 3.96 | 3.94     | 3.93  | 3.96     | 3.98  | 3.99    | 3.93      | 3.96 |
> | necessity-only | 2.58 | 2.68     | 2.58  | 2.83     | 2.95  | 3.31    | 2.84      | 2.82 |
> | both           | 2.37 | 2.39     | 2.37  | 2.85     | 3.35  | 3.23    | 2.71      | 2.75 |
>
> ---
>
> ### On Adapting TRM to Tool Types with More Complex Utility ---> Question 2
> On the one hand, we acknowledge that our current decomposition of tool utility into necessity and quality is relatively simple, and certain tools may require more nuanced, task-specific definitions of utility (e.g., graded usefulness, cost-aware or multi-objective rewards, or tools producing rich intermediate outputs). Adapting TRM to these settings would likely benefit from customized reward design to reflect the specific characteristics of the tool.
>
> On the other hand, the two dimensions we consider are relatively general: necessity captures whether a tool call contributes substantive progress toward task completion, and quality captures whether the tool is invoked with reasonable parameters or used correctly. Our prior results (as discussed in response to Weakness 1) further support this generality, showing that TRM improves performance across diverse multi-tool, multi-task scenarios beyond search and code tools. This suggests that, while more complex tools may require richer or more tailored reward definitions, the TRM construction pipeline can potentially be adapted to a wide range of tool types.
>
> We have clarified this limitation in lines 832-844 of our revised manuscript and note that extending TRM to richer or more complex utility formulations is an important direction for future work.

---

> ### Author Response · Authors · 2025-11-23
> **Response to Reviewer R2Sf (part 3/3)**
>
> ### On Failure Cases and Process-Level Transparency of TRM ---> Question 3
> To ensure the reliability and transparency of TRM, we first validated our training data for necessity and quality labels. As reported in Table 5, the dataset achieves relatively high accuracy on both necessity and quality, indicating that the annotated rewards are largely correct and reliable.
>
> **Table 5: TRM Training Data Quality Evaluation (Accuracy)**
> | Evaluation Method | Necessity (%) | Quality (%) |
> | ----------------- | ------------- | ----------- |
> | Human             | 89            | 89          |
>
>
> In addition to quantitative validation, we have conducted detailed case analyses of TRM predictions. These analyses reveal insights into failure cases and the distinctions between the two evaluation dimensions:
> * cases with low necessity: Tool calls are unnecessary, often because the problem is trivial, information is already available in context, or the tool is called redundantly. For example, for the question `The product 8 × .25 × 2 × .125 =`, invoking a code tool is unnecessary, as the computation can be done directly. TRM appropriately assigns a low necessity score in such scenarios.
> * cases with low quality: Tool calls usage is flawed, e.g., the query or parameters were incorrect, incomplete, or inefficient. For example, for the question `Joachim Gottschalk, was a German stage and film actor during the late 1930s, a romantic lead in the style of who, which was an English stage and film actor, director and producer?`, invoking a search tool with query `Joachim Gottschalk acting style comparison` omitted critical elements (e.g., specifying “English actor” and the name-retrieval target), which may lead to irrelevant or incomplete results. In this case, TRM appropriately assigns a low quality score.
>
> More cases are shown in our updated manuscript (Table 8).
>
> ---
>
> ### Scope of TRM and Relation to Prior Work (PAL and StableToolBench) ---> Weakness 3
> We have supplemented our experiments with PAL as suggested (lines 385-386 in our revised manuscript). Regarding StableToolBench, it primarily focuses on open-domain, open-ended tasks, which differ from the more structured, verifiable task setting studied in our work. Our method, TRM, is designed for scenarios where the final outcome can be reliably verified, allowing per-tool-call rewards to provide finer-grained learning signals. In contrast, applying TRM to fully open-ended task, where the correctness of the final outcome is difficult to verify, would require additional considerations and is outside the scope of this paper. We explicitly acknowledge this limitation in the revised manuscript (lines 832-844).
>
> ---
>
> ### Clarification on the Use of the Term “Reward Model” vs. “Reward Function” ---> Minor issue
> Thanks for pointing out the potential confusion. The confusion likely comes from the fact that the training labels for our Tool-call Reward Model (TRM) are defined along two specific dimensions, necessity and quality.
>
> To clarify our terminology:
> * A reward function means that the labels are computed by fixed, deterministic rules.
> * A reward model means that the reward signal is predicted by a model.
>
> In our case, the signals for necessity and quality are not obtained by hand-crafted rules; they are *predicted by a model and used as labels to train our TRM*. Moreover, these two dimensions are relatively general, and our new extended experiments on more tools (response to weakness 1) further support their generality. Therefore, we adopt the term “reward model” for this component. If necessary, we are also happy to follow your suggestion in next version.

---

> ### Author Response · Authors · 2025-11-27
> **Inquiry About Discussion Status**
>
> Dear Reviewer R2Sf,
>
> Thank you again for your time and for the constructive comments on our paper. We fully appreciate that everyone has many competing commitments, and we are grateful for your efforts. As the discussion deadline is approaching, we just wanted to briefly follow up. To address your concerns, we have conducted additional experiments and provided further clarifications to the best of our ability.
>
> If you have any remaining questions or would like to discuss any point in more detail, we would be very happy to continue the discussion.
>
> Best regards,
>
> Team 22507

---

### Official Review · Reviewer_S77r · 2025-10-31

**Soundness:** 2
**Presentation:** 2
**Contribution:** 2
**Rating:** 6
**Confidence:** 4

**Summary:**

The paper introduces a Tool Reward Model (TRM) that scores each tool call with a binary “necessity × quality” signal and combines these turn-level scores with an outcome reward for the final answer. TRM is integrated into PPO/GRPO via turn-level credit assignment and is used for RL training.

**Strengths:**

1. The writing is clear.
2. Extensive experiments validate the effectiveness of TRM.

**Weaknesses:**

1. Necessity/quality labels are auto-judged by the same model family that produced the rollouts, no multi-judge or human verification is reported, which may imprint model bias.
2. Tool coverage is narrow (search, code), generality to other tool families is unclear.
3. There is no direct comparison to current PRM/agent-PRM baselines.

**Questions:**

1. How do necessity and quality contribute individually? A comparison among (i) necessity-only, (ii) quality-only, and (iii) the combined  reward (as in the paper) would be helpful.

2. How do you ensure the distillation labels remain reliable for long rollouts, given that feeding entire traces to a judge model may introduce unfaithful judgments or evaluation bias?

3. Beyond search and code, how well does TRM generalize to other tools?

4. How does TRM size affect downstream results? Does a scaling trend emerge for reward-model size?

5. Adding a comparison to existing PRM/agent-PRM baselines such as [1] would further strengthen the evaluation.

[1] Process Reward Models for LLM Agents: Practical Framework and Directions.

---

> ### Author Response · Authors · 2025-11-23
> **Response to Reviewer S77r (part 1/3)**
>
> We sincerely appreciate your efforts and the valuable feedback on our work.
>
> ---
>
> ### TRM Training Data Validation ---> Weakness 1
> We appreciate the concern regarding potential model bias when necessity and quality labels are judged by the same model family that produced the rollouts. To address this, we validated our TRM training data using both human annotators and a multi-model judge (DeepSeek-R1, Qwen2.5-72B-Instruct, and GPT-4o-mini).
>
> As shown in Table 1, the dataset achieves relatively high accuracy on both necessity (Human: 89%, Multi-model: 86%) and quality (Human: 89%, Multi-model: 75%), indicating that the annotated rewards are largely correct and reliable. We also plan to open-source all code and data to provide full transparency and facilitate further research in this area.
>
> Relevant results have been updated in lines 1420–1431 of the revised manuscript.
>
> **Table 1: TRM Training Data Quality Evaluation (Accuracy)**
> | Evaluation Method | Necessity (%) | Quality (%) |
> | ----------------- | ------------- | ----------- |
> | Human             | 89            | 89          |
> | Multi-model       | 86            | 75          |
>
> ---
>
> ### Tool Coverage and Generality ---> Weakness 2; Question 3
> To further evaluate the generality of our approach beyond search and code tools, we consider the multi-tool scenarios introduced in ReCall [1]. These scenarios include tasks such as library management, employee management, and travel planning, which require the use of multiple APIs (e.g., searching and reserving books, calculating salaries, updating flight bookings, etc.). These additional experiments (see Table 2) show that introducing TRM continues to provide consistent improvements in broader multi-tool settings.
>
> Relevant results have been updated in lines 1432–1444 of the revised manuscript.
>
> **Table 2: Performance in More Diverse Multi-Tool Scenarios on Qwen2.5-7B-Instruct**
> | Method                       | F1    |
> | ---------------------------- | ----- |
> | Qwen2.5-7B-Instruct          | 7.37  |
> | ReCall (outcome-reward-only) | 39.71 |
> | ReCall + TRM                 | 43.28 |
>
> [1] https://attractive-almandine-935.notion.site/ReCall-Learning-to-Reason-with-Tool-Call-for-LLMs-via-Reinforcement-Learning-1d7aec91e9bb8006ad40f9edbfe2191a

---

> ### Author Response · Authors · 2025-11-23
> **Response to Reviewer S77r (part 2/3)**
>
> ### Comparison with Process-Supervised Tool-Use Methods ---> Weakness 3; Question 5
> First, we include a baseline called StepSearch [2]. StepSearch is specifically designed for search-based QA scenarios and evaluates intermediate search tool usage from two perspectives:
> * *Query relevance*: whether the search query is reasonable
> * *Information gain*: whether the retrieved documents provide additional information useful for answering the question.
>
> The procedure is as follows:
> * Given a question, an LLM first generates search queries, and a search engine retrieves documents based on these queries. The resulting queries and documents are treated as the ground-truth search interactions for answering the question.
> * Query relevance is computed as the maximum word-level F1 score between the model-generated queries and the ground-truth queries.
> * Information gain is measured by checking whether the retrieved documents are more similar to the ground-truth documents than the documents retrieved in previous searches.
>
> Additionally, we also compare TRM with AgentPRM [3], a general process-supervised method that is not specifically designed for search-based QA. For AgentPRM, each tool call is labeled as 1 if, starting from this call, an LLM can reach a correct final answer in at least one of multiple rollouts; otherwise the label is 0. This gives a relatively strong but still generic process supervision signal at the per-call level.
>
> Table 3 shows that TRM consistently outperforms both StepSearch and AgentPRM across all seven QA benchmarks. StepSearch, although tailored to search QA, yields inconsistent gains and even hurts the average performance compared to the outcome-only baseline Search-R1 [4]. AgentPRM provides some improvements over Search-R1 on several datasets but still lags behind TRM in both individual benchmarks and overall average. These results indicate that generic process supervision signals (AgentPRM) and hand-crafted search-specific signals (StepSearch) are both less reliable than our per-tool-call reward modeling, and they empirically support the effectiveness of TRM.
>
> Relevant results have been updated in lines 429–469 of the revised manuscript.
>
> **Table 3: TRM vs. Process-Supervised Tool-Use Method (StepSearch) on Qwen2.5-3B-Instruct**
> | Method                 | NQ    | TriviaQA | PopQA | HotpotQA | 2wiki | Musique | Bamboogle | Avg.  |
> | ---------------------- | ----- | -------- | ----- | -------- | ----- | ------- | --------- | ----- |
> | Search-R1              | 36.93 | 54.48    | 35.85 | 32.65    | 32.47 | 12.08   | 24.80     | 32.75 |
> | Search-R1 + StepSearch | 37.53 | 55.17    | 39.20 | 29.75    | 27.65 | 7.74    | 17.60     | 30.66 |
> | Search-R1 + AgentPRM   | 38.01 | 54.62    | 37.07 | 32.78    | 31.15 | 10.22   | 19.20     | 31.86 |
> | Search-R1 + TRM (ours) | 39.58 | 57.78    | 40.61 | 34.80    | 33.22 | 12.91   | 25.60     | 34.93 |
>
> [2] Wang, Ziliang, et al. "StepSearch: Igniting LLMs Search Ability via Step-Wise Proximal Policy Optimization." arXiv preprint arXiv:2505.15107 (2025).
>
> [3] Choudhury S. Process reward models for llm agents: Practical framework and directions[J]. arXiv preprint arXiv:2502.10325, 2025.
>
> [4] Bowen Jin, Hansi Zeng, Zhenrui Yue, Jinsung Yoon, Sercan O Arik, Dong Wang, Hamed Zamani, and Jiawei Han. Search-r1: Training LLMs to reason and leverage search engines with reinforcement learning. In Second Conference on Language Modeling, 2025.

---

> ### Author Response · Authors · 2025-11-23
> **Response to Reviewer S77r (part 3/3)**
>
> ### Ablation on the Necessity and Quality of Tool Calls ---> Question 1
> We acknowledge this insightful suggestion. Table 4 shows that using quality-only yields the lowest model performance. Table 5 further reveals that relying solely on quality leads to overuse of tools, which negatively impacts performance. In contrast, using necessity-only reduces the number of tool calls but may compromise the quality of each call, also limiting overall performance. Combining both necessity and quality achieves the best overall performance while maintaining a relatively stable number of tool calls across datasets.
>
> Relevant results have been updated in lines 480–485 of the revised manuscript.
>
> **Table 4: Model Performance Ablation on Tool-Call Necessity and Quality (Qwen2.5-3B-Instruct)**
> | Method         | NQ    | TriviaQA | PopQA | HotpotQA | 2wiki | Musique | Bamboogle | Avg.  |
> | -------------- | ----- | -------- | ----- | -------- | ----- | ------- | --------- | ----- |
> | quality-only   | 38.95 | 56.17    | 38.94 | 31.06    | 27.93 | 8.44    | 16.00     | 31.07 |
> | necessity-only | 37.42 | 54.06    | 37.21 | 32.46    | 31.80 | 11.46   | 21.60     | 32.29 |
> | both           | 39.58 | 57.78    | 40.61 | 34.80    | 33.22 | 12.91   | 25.60     | 34.93 |
>
> **Table 5: Tool-Call Number Ablation on Necessity and Quality (Qwen2.5-3B-Instruct)**
> | Method         | NQ   | TriviaQA | PopQA | HotpotQA | 2wiki | Musique | Bamboogle | Avg. |
> | -------------- | ---- | -------- | ----- | -------- | ----- | ------- | --------- | ---- |
> | quality-only   | 3.96 | 3.94     | 3.93  | 3.96     | 3.98  | 3.99    | 3.93      | 3.96 |
> | necessity-only | 2.58 | 2.68     | 2.58  | 2.83     | 2.95  | 3.31    | 2.84      | 2.82 |
> | both           | 2.37 | 2.39     | 2.37  | 2.85     | 3.35  | 3.23    | 2.71      | 2.75 |
>
>
>
> ---
>
>
> ### Reliability of Distillation Labels for Long Rollouts ---> Question 2
> Thank you for raising this important question. In practice, we follow prior work [4] on tool-augmented QA and truncate search results and trajectories to keep rollout length within a reasonable range (~8K), which reduces the risk of extremely long traces that could heavily amplify judge bias.
>
> More importantly, the main focus of our work is not on handling very long rollouts, but on addressing a different limitation of outcome-only rewards: they tend to penalize all tool calls in an incorrect trajectory uniformly, even when some tool calls are actually reasonable. TRM is introduced precisely to provide additional fine-grained per-tool-call supervision on top of outcome-only feedback, thereby mitigating this gradient conflict and improving the quality of tool-use decisions.
>
> We agree that the reliability of process supervision for very long rollouts is an important open problem for tool-use research in general, and our current work does not fully resolve it. We explicitly acknowledge this as a limitation and leave a more systematic treatment of long-rollout evaluation and distillation as an interesting direction for future work (lines 832-844 in our revised manuscript).
>
> ---
>
> ### Effect of TRM Size and Scaling Trend ---> Question 4
> We do observe a scaling trend that is broadly consistent with the intuition that larger reward models can provide better supervision. As shown in Figure 3(a) of the revised manuscript, increasing TRM size generally leads to improved downstream QA performance when sufficient TRM training data are available.
>
> However, we also find that this trend is not purely monotonic. When the amount of distillation data is relatively limited, overly large TRMs are more prone to overfitting, which can in turn hurt downstream performance. In other words, the benefits of scaling up TRM capacity depend on having enough high-quality training trajectories; otherwise, the larger model may fit noise in the labels rather than providing more reliable rewards.
>
> [4] Bowen Jin, Hansi Zeng, Zhenrui Yue, Jinsung Yoon, Sercan O Arik, Dong Wang, Hamed Zamani, and Jiawei Han. Search-r1: Training LLMs to reason and leverage search engines with reinforcement learning. In Second Conference on Language Modeling, 2025.

---

> ### Author Response · Authors · 2025-11-27
> **Inquiry About Discussion Status**
>
> Dear Reviewer S77r,
>
> Thank you again for your time and for the constructive comments on our paper. We fully appreciate that everyone has many competing commitments, and we are grateful for your efforts. As the discussion deadline is approaching, we just wanted to briefly follow up. To address your concerns, we have conducted additional experiments and provided further clarifications to the best of our ability.
>
> If you have any remaining questions or would like to discuss any point in more detail, we would be very happy to continue the discussion.
>
> Best regards,
>
> Team 22507

---

### Official Review · Reviewer_8EFK · 2025-11-01

**Soundness:** 3
**Presentation:** 3
**Contribution:** 2
**Rating:** 6
**Confidence:** 2

**Summary:**

This paper proposes the Tool-call Reward Model (TRM), a specialized process reward model designed to evaluate and reward tool invocations. The authors further enhance TRM's application in PPO and GRPO training through refined credit assignment and turn-level advantage estimation. The effectiveness of the proposed method is demonstrated on search-based QA and Python code-based mathematical tasks.

**Strengths:**

1. Current tool-calling agent training predominantly relies on outcome rewards. The proposed TRM addresses a crucial need for process-level evaluation in tool-calling agent training, which is highly valuable for the field.
2. The introduction in this paper is detailed, with comprehensive explanations of both the background and methodology.
3. The authors conduct comprehensive experiments, including both scaling assessments of TRM and practical evaluations on search-based QA and coding tasks. The experimental results demonstrate consistent and substantial improvements over baseline methods across different tasks.

**Weaknesses:**

1. The paper lacks critical details regarding the construction of TRM training data. Specifically, how do the authors ensure the correctness of annotated tool call rewards? Evaluating the correctness and necessity of tool invocations at each turn requires a comprehensive understanding of the overall problem and reasoning process, which poses significant challenges for models, particularly when tools provide information beyond the model's knowledge scope. Have the authors conducted any validation of the annotation quality?
2. For open-ended tasks such as search-based QA, it is challenging to directly assess the effectiveness of final results. The paper does not adequately address how such evaluations are conducted, which raises concerns about the reliability of the reported improvements.

**Questions:**

N/A

---

> ### Author Response · Authors · 2025-11-23
> **TRM Training Data Quality ---> Weakness 1 (part 1/2)**
>
> We sincerely thank you for recognizing our work and for providing your valuable feedback.
>
> ---
>
> We appreciate the emphasis on validating the correctness of annotated tool-call rewards. We have validated our TRM training data in two ways.
>
> ---
>
> ### Human Evaluation and Multi-model Judge
> First, we directly evaluate the quality of our TRM training data by randomly sampling 100 trajectories and assessing their tool-call annotations using both human annotators and a multi-model judge (DeepSeek-R1, Qwen2.5-72B-Instruct, and GPT-4o-mini). Importantly, the judge (human or model) is given the *complete trajectory*, including the original question, the full reasoning process, all tool invocations, and tool outputs. This ensures that necessity and correctness are evaluated with a global view of the problem and reasoning process, rather than based on isolated tool calls. This setup also mitigates the concern that tools may return information beyond the base model’s knowledge scope, since the judge directly observes the tool outputs themselves rather than relying solely on prior model knowledge.
>
> As shown in Table 1, the dataset achieves relatively high accuracy on both necessity (Human: 89%, Multi-model: 86%) and quality (Human: 89%, Multi-model: 75%), indicating that the annotated rewards are largely correct. We also plan to open-source all code and data to provide full transparency and facilitate further research in this area.
>
> Relevant results have been updated in lines 1420–1431 of the revised manuscript.
>
> **Table 1: TRM Training Data Quality Evaluation (Accuracy)**
> | Evaluation Method | Necessity (%) | Quality (%) |
> | ----------------- | ------------- | ----------- |
> | Human             | 89            | 89          |
> | Multi-model       | 86            | 75          |
>
> ---
>
> ### Ablation Study on the Necessity and Quality of Tool Calls
> Second, we perform an ablation study on the necessity and quality signals to verify that these annotations are not only self-consistent but also functionally important. Table 2 shows that using quality-only yields the lowest model performance. Table 3 further reveals that relying solely on quality leads to overuse of tools, which negatively impacts performance. In contrast, using necessity-only reduces the number of tool calls but may compromise the quality of each call, also limiting overall performance. Combining both necessity and quality achieves the best overall performance while maintaining a relatively stable number of tool calls across datasets. This demonstrates that our annotated signals are meaningful and critical for model effectiveness.
>
> Relevant results have been updated in lines 480–485 of the revised manuscript.
>
> **Table 2: Model Performance Ablation on Tool-Call Necessity and Quality (Qwen2.5-3B-Instruct)**
> | Method         | NQ    | TriviaQA | PopQA | HotpotQA | 2wiki | Musique | Bamboogle | Avg.  |
> | -------------- | ----- | -------- | ----- | -------- | ----- | ------- | --------- | ----- |
> | quality-only   | 38.95 | 56.17    | 38.94 | 31.06    | 27.93 | 8.44    | 16.00     | 31.07 |
> | necessity-only | 37.42 | 54.06    | 37.21 | 32.46    | 31.80 | 11.46   | 21.60     | 32.29 |
> | both           | 39.58 | 57.78    | 40.61 | 34.80    | 33.22 | 12.91   | 25.60     | 34.93 |
>
> **Table 3: Tool-Call Number Ablation on Necessity and Quality (Qwen2.5-3B-Instruct)**
> | Method         | NQ   | TriviaQA | PopQA | HotpotQA | 2wiki | Musique | Bamboogle | Avg. |
> | -------------- | ---- | -------- | ----- | -------- | ----- | ------- | --------- | ---- |
> | quality-only   | 3.96 | 3.94     | 3.93  | 3.96     | 3.98  | 3.99    | 3.93      | 3.96 |
> | necessity-only | 2.58 | 2.68     | 2.58  | 2.83     | 2.95  | 3.31    | 2.84      | 2.82 |
> | both           | 2.37 | 2.39     | 2.37  | 2.85     | 3.35  | 3.23    | 2.71      | 2.75 |

---

> ### Author Response · Authors · 2025-11-23
> **Scope and Open-Ended Tasks ---> Weakness 2 (part 2/2)**
>
> We agree that, for open-ended tasks such as search-based QA, reliably validating the final outcome is intrinsically challenging. In such open-domain settings, RL based purely on outcome rewards is still in an early stage, precisely because the correctness of the final answer often cannot be robustly verified [1]. In contrast, our work deliberately focuses on a setting where the final outcome is verifiable. Even in this more favorable regime, pure outcome-based RL still exhibits a clear limitation: when the final answer is incorrect, the entire trajectory is penalized uniformly, even if many intermediate tool calls are actually correct and useful. TRM is designed to address this issue by providing fine-grained, per-tool-call rewards on top of an outcome-based RL framework. As such, fully open-ended tasks, where the final outcome cannot be reliably verified, are outside the scope of this work, and we explicitly acknowledge this limitation in the paper (lines 833-844 in our revised manuscript).
>
> Furthermore, we understand that this concern may also reflect an interest in the generality of our approach beyond the specific benchmarks we report. To this end, we follow the ReCall [2] work to evaluate our method in more diverse multi-tool scenarios. These scenarios include tasks such as library management, employee management, and travel planning, which require the use of multiple APIs (e.g., searching and reserving books, calculating salaries, updating flight bookings, etc.). These additional experiments (see Table 4) show that introducing TRM continues to provide consistent improvements in broader multi-tool settings. Additional details can be found in lines 1432-1444 in our revised manuscript.
>
> **Table 4: Performance in More Diverse Multi-Tool Scenarios on Qwen2.5-7B-Instruct**
> | Method                       | F1    |
> | ---------------------------- | ----- |
> | Qwen2.5-7B-Instruct          | 7.37  |
> | ReCall (outcome-reward-only) | 39.71 |
> | ReCall + TRM                 | 43.28 |
>
> [1] Gunjal A, Wang A, Lau E, et al. Rubrics as rewards: Reinforcement learning beyond verifiable domains[J]. arXiv preprint arXiv:2507.17746, 2025.
>
> [2] https://attractive-almandine-935.notion.site/ReCall-Learning-to-Reason-with-Tool-Call-for-LLMs-via-Reinforcement-Learning-1d7aec91e9bb8006ad40f9edbfe2191a

---

> ### Author Response · Authors · 2025-11-27
> **Inquiry About Discussion Status**
>
> Dear Reviewer 8EFK,
>
> Thank you again for your time and for the constructive comments on our paper. We fully appreciate that everyone has many competing commitments, and we are grateful for your efforts. As the discussion deadline is approaching, we just wanted to briefly follow up. To address your concerns, we have conducted additional experiments and provided further clarifications to the best of our ability.
>
> If you have any remaining questions or would like to discuss any point in more detail, we would be very happy to continue the discussion.
>
> Best regards,
>
> Team 22507

---

### Official Review · Reviewer_TgrD · 2025-11-01

**Soundness:** 2
**Presentation:** 3
**Contribution:** 2
**Rating:** 2
**Confidence:** 5

**Summary:**

This paper proposes a Tool-call Reward Model (TRM), a process reward model and a training/credit-assignment recipe to integrate TRM with PPO/GRPO. The authors distill per-turn "necessity" and "quality" labels from tool-enabled rollouts, train a binary classifier head on top of a small LLM (1.5B–7B). Experiments on search-based QA and code-based math claim consistent gains over outcome-only RL baselines.

**Strengths:**

1. The motivation is straightforward and easy to follow. It shows consistent improvements on search-QA and code-based math.

2. The paper identifies the limitation of outcome-only rewards for agentic tool use and motivates per-call supervision with concrete failure modes.

3. It finds that mid-sized TRMs (1.5B–3B) trained on ~10k labeled trajectories suffice, with larger models overfitting at this data scale

**Weaknesses:**

1. Insufficient comparison with existing tool-augmented LLMs. It omits direct comparisons to process-supervised tool-use methods that score steps or calls (e.g., rule/PRM-guided selection for search or code).

2. Missing positioning relative to tool-based RM in the literature. The paper does not cite or discuss the line of work referred to as tool-augmented reward modeling [1]. The motivation of proposing Tool-call Reward Model is unclear. The authors did not discuss with previous tool-augmented RM and completely ignores previous literature.

3. While turn-level estimation mitigates "fewer calls is better", the paper does not report behavioral metrics, such as average #tool-calls, redundant calls, early-stop rates, or task-time/latency impacts under TRM vs. baselines.

4. TRM inference is performed during RL and optionally at inference (best-of-n). The paper should quantify the compute/memory overheads.

5. Lack sufficient ablations. Ablations to disambiguate distillation vs. TRM:
   - Train a trajectory-level RM/PRM (no per-call labels) on the same data, and compare to TRM under equal compute.
   - Train a tool-based RM such as [1] on the same data, to demonstrate the good claims of PRM. The paper states TRM is a specific PRM for tool invocation, yet it does not compare to a generic PRM trained on the same trajectories. At minimum, the authors should implement a standard PRM that scores intermediate steps or turns

6. The training data is completely generated by LLMs, without mentioning any interventions or pipelines. The author did not report the details of distilled data.

7. It is unclear how the authors evaluate Tool-call RM on downstream tasks, such as AIME.

8. The authors evaluate TRM on best-of-n inference. The author should report the comparisons with previous reward models. Additionally, the author may report the RL training performance using proposed TRM.

9. The usage of "necessity" and "quality" requires a detailed ablation comparison, while the paper only describes the concept/motivation in the method section. Most importantly, and how to do the quality check, how to measure the ground truth score of the intermediate steps, and how does this impact on the final results, remains still unclear.



References:

[1] tool-augmented reward modeling. ICLR 2024.

**Questions:**

See above

---

> ### Author Response · Authors · 2025-11-23
> **Clarification of Certain Misunderstandings (part 1/4)**
>
> We are grateful for your thorough and insightful comments, which have greatly helped us improve our work.
>
> ---
>
> ### Motivation Behind the Tool-call Reward Model ---> Weakness 2 and 5.2
> **Motivation (weakness 2)** Our work focuses on *improving the ability of large language models to use external tools*. Recently, most reinforcement learning methods that rely *solely on outcome-based rewards* have become popular for training these models. However, such methods can suffer from gradient conflicts [1], meaning that if a trajectory results in an incorrect final answer, the model may receive negative signals for all tool calls in that trajectory, even when some intermediate tool calls are correct. This *indiscriminate penalization* can lead to suboptimal model performance. To mitigate this issue, we propose the Tool-call Reward Model (TRM), which *evaluates each tool call within a complete trajectory and provides additional reward signals to complement the outcome reward*. This design ensures that correct tool calls within a trajectory receive positive reinforcement, even if the final outcome is incorrect.
>
> **Distinction from Tool-augmented Reward Model [2] (weakness 2)** The tool-call reward model evaluates the quality of each tool call made by a model, providing a separate score for every tool invocation within a trajectory. For example, if a trajectory involves three tool calls, the tool-call reward model produces three individual scores. In contrast, a tool-augmented reward model allows the reward model itself to call tools during evaluation but produces a single score for the entire trajectory. Thus, *while tool-call reward model focuses on fine-grained assessment of each tool call, tool-augmented reward model focuses on enhancing reward estimation by permitting tool usage during evaluation*. This distinction is substantiated by the abstract of [2], which states:
> > While conventional reward models (RMs) have exhibited remarkable scalability, they oft struggle with fundamental functionality such as arithmetic computation, code execution, and factual lookup. In this paper, we propose a tool-augmented preference modeling approach, named Themis, to address these limitations by **empowering RMs with access to external environments, including calculators and search engines**.
>
> **Connection to Process Reward Model (weakness 5.2)** The TRM can be viewed as a form of Process Reward Model (PRM), as it takes a complete trajectory as input and outputs an evaluation for each tool call within that trajectory. However, it *differs from commonly used PRM approaches in its focus*. Traditional PRM methods typically concentrate on reasoning tasks, such as mathematical problem solving, whereas TRM specifically focuses on tool-use behavior in large language models. In this sense, **our TRM instantiates exactly the kind of tool-based PRM that was requested in the review**.
>
> Additional details can be found in the second and third paragraphs of the Introduction (lines 042–081) and Figure 1 of the manuscript.
>
> ---
>
> ### Evaluation of TRM ---> Weakness 7 and 8
> In this work, we evaluate the TRM in two different settings:
> * *Best-of-N (BoN) inference* [1,3]: TRM is used as a verifier to select the best answer among multiple model outputs. Specifically, for a given prompt, the model generates N trajectories, and the score of each trajectory is computed as the product of the scores of all tool calls within that trajectory [1]. Experimental results show that, as N increases, the performance of BoN with TRM as the verifier continues to improve, and it consistently outperforms common verifiers such as majority vote [4]. Further details can be found in the manuscript (lines 291–297), and the experimental results are presented in Figure 3.
> * *RL training*: TRM is incorporated into the RL training process to provide fine-grained reward signals for tool usage, guiding the model toward better tool-call behavior. Experimental results show that incorporating TRM leads to better performance than RL methods that rely solely on outcome rewards. Additional details are provided in Section 2.3 of the manuscript, and the experimental results are reported in Tables 1 and 2.
>
> [1] Hunter Lightman, et al. Let’s verify step by step. In The Twelfth International Conference on Learning Representations, 2024.
>
> [2] Li, Lei, et al. "Tool-augmented reward modeling." In The Twelfth International Conference on Learning Representations, 2024.
>
> [3] Liangchen Luo, et al. Improve mathematical reasoning in language models with automated process supervision, 2025
>
> [4] Xuezhi Wang, et al. Self-Consistency Improves Chain of Thought Reasoning in Language Models. In The Eleventh International Conference on Learning Representations, 2023.

---

> ### Author Response · Authors · 2025-11-23
> **Supplementary Experimental Results (part 2/4)**
>
> ### Direct Comparison with Process-Supervised Tool-Use Methods ---> Weakness 1
> Thanks for the great suggestion to compare with other process-supervised tool-use methods. To address this, we include a baseline called StepSearch [1]. StepSearch is specifically designed for search-based QA scenarios and evaluates intermediate search tool usage from two perspectives:
> * *Query relevance*: whether the search query is reasonable
> * *Information gain*: whether the retrieved documents provide additional information useful for answering the question.
>
> The procedure is as follows:
> * Given a question, an LLM first generates search queries, and a search engine retrieves documents based on these queries. The resulting queries and documents are treated as the ground-truth search interactions for answering the question.
> * Query relevance is computed as the maximum word-level F1 score between the model-generated queries and the ground-truth queries.
> * Information gain is measured by checking whether the retrieved documents are more similar to the ground-truth documents than the documents retrieved in previous searches.
>
> Additionally, we also compare TRM with AgentPRM [2], a general process-supervised method that is not specifically designed for search-based QA. For AgentPRM, each tool call is labeled as 1 if, starting from this call, an LLM can reach a correct final answer in at least one of multiple rollouts; otherwise the label is 0. This gives a relatively strong but still generic process supervision signal at the per-call level.
>
> Table 1 shows that TRM consistently outperforms both StepSearch and AgentPRM across all seven QA benchmarks. StepSearch, although tailored to search QA, yields inconsistent gains and even hurts the average performance compared to the outcome-only baseline Search-R1 [3]. AgentPRM provides some improvements over Search-R1 on several datasets but still lags behind TRM in both individual benchmarks and overall average. These results indicate that generic process supervision signals (AgentPRM) and hand-crafted search-specific signals (StepSearch) are both less reliable than our per-tool-call reward modeling, and they empirically support the effectiveness of TRM.
>
> Relevant results have been updated in lines 429–469 of the revised manuscript.
>
> **Table 1: TRM vs. Process-Supervised Tool-Use Method (StepSearch) on Qwen2.5-3B-Instruct**
> | Method                 | NQ    | TriviaQA | PopQA | HotpotQA | 2wiki | Musique | Bamboogle | Avg.  |
> | ---------------------- | ----- | -------- | ----- | -------- | ----- | ------- | --------- | ----- |
> | Search-R1              | 36.93 | 54.48    | 35.85 | 32.65    | 32.47 | 12.08   | 24.80     | 32.75 |
> | Search-R1 + StepSearch | 37.53 | 55.17    | 39.20 | 29.75    | 27.65 | 7.74    | 17.60     | 30.66 |
> | Search-R1 + AgentPRM   | 38.01 | 54.62    | 37.07 | 32.78    | 31.15 | 10.22   | 19.20     | 31.86 |
> | Search-R1 + TRM (ours) | 39.58 | 57.78    | 40.61 | 34.80    | 33.22 | 12.91   | 25.60     | 34.93 |
>
> ---
>
> ### Tool-call Number Comparison: Turn-level vs. Group-level ---> Weakness 3
> We first recall the distinction between turn-level and group-level advantage estimation in GRPO. In the group-level baseline, advantage estimation is performed collectively over all tool-call rewards within a group, whereas in turn-level estimation, tool calls are divided into sequential groups (e.g., the first tool calls form the first group, the second calls form the second group, etc.), with advantages computed separately for each group.
>
> In the original manuscript, we reported the trend of tool-call numbers during training under these two methods (Figure 7). Compared with turn-level estimation, group-level tends to produce fewer tool calls, consistent with the “fewer calls is better” reward hacking bias. To further validate this, we report tool-call numbers on evaluation benchmarks in Table 2. The results confirm that group-level estimation generally leads to fewer tool calls than turn-level estimation, consistent with the observed training trend.
>
> Relevant results have been updated in lines 1404-1411 of the revised manuscript.
>
> **Table 2: Average Tool-Call Numbers on Evaluation Benchmarks: Turn-level vs. Group-level**
> | Method      | NQ   | TriviaQA | PopQA | HotpotQA | 2wiki | Musique | Bamboogle | Avg. |
> | ----------- | ---- | -------- | ----- | -------- | ----- | ------- | --------- | ---- |
> | Group-level | 1.99 | 1.93     | 2.00  | 2.42     | 2.81  | 3.01    | 2.37      | 2.36 |
> | Turn-level  | 2.59 | 2.44     | 2.48  | 2.68     | 3.01  | 3.19    | 2.68      | 2.72 |
>
> [1] Wang, Ziliang, et al. "StepSearch: Igniting LLMs Search Ability via Step-Wise Proximal Policy Optimization." arXiv preprint arXiv:2505.15107 (2025).
>
> [2] Choudhury S. Process reward models for llm agents: Practical framework and directions[J]. arXiv preprint arXiv:2502.10325, 2025.

---

> ### Author Response · Authors · 2025-11-23
> **Supplementary Experimental Results (part 3/4)**
>
> ### Resource Overhead Introduced by TRM ---> Weakness 4
> As shown in Table 3, incorporating TRM introduces only an 8.8% overhead per training step, indicating minimal additional compute cost. BoN inference experiences a 50% increase in per-sample time with TRM; however, the absolute time remains small (with the full BoN evaluation taking ~20 minutes), which is practically negligible.
>
> **Table 3: Training and BoN Inference Speed with vs. without TRM on Qwen2.5-3B-Instruct with 8xA800 GPUs**
> | Method  | Training (s/step) | BoN Inference (s/sample) |
> | ------- | ----------------- | ------------------------ |
> | w/o TRM | 56.9              | 0.14                     |
> | w/ TRM  | 61.9  (+8.8%)     | 0.21    (+50%)           |
>
> In terms of memory, incorporating TRM does not introduce noticeable overhead. This is because TRM is generally smaller than the policy model and is invoked serially during both training and BoN inference. During training, TRM scores rollouts before the policy update, and during inference, trajectories are first collected by the policy model and then scored by TRM. As a result, TRM and the policy model are never resident in memory simultaneously, and since TRM is smaller than the policy model, the peak memory usage remains largely unchanged compared to running the policy model alone.
>
> Overall, these results demonstrate that TRM can be integrated with modest computational and memory cost during both training and BoN evaluation. Relevant results have been updated in lines 1398–1402 of the revised manuscript.
>
> ---
>
> ### Ablation Study to Disambiguate Distillation and TRM ---> Weakness 5.1
> We greatly appreciate this constructive suggestion. To disambiguate the effects of distillation and TRM, we introduce two additional baselines:
> * *Outcome Reward Model (ORM)*: We train an ORM on the same data used to train TRM. ORM provides a single score for each entire trajectory, which is then combined with answer correctness via a weighted sum.
> * *TRM as a verifier*: We use the trained TRM to score trajectories, where the geometric mean of all per-tool-call scores along a trajectory is combined with answer correctness.
>
> Both baselines leverage the distillation data; the key distinction is that TRM provides fine-grained, per-tool-call evaluation signals, allowing us to clearly separate the contributions of distillation from those of TRM.
>
> Table 4 highlights the effects of distillation and TRM. Using a trajectory-level ORM (Search-R1-ORM) underperforms even the answer-only baseline (Search-R1 [3]), likely because ORM introduces additional noise from answer evaluation. Using TRM as a verifier (Search-R1-TRM-verifier) yields some improvement over answer-only evaluation, but still falls short of the full TRM model. These results demonstrate that the fine-grained per-tool-call signals provided by TRM are crucial for its effectiveness, clearly distinguishing its contribution from that of distillation.
>
> Relevant results have been updated in lines 471–477 of the revised manuscript.
>
> **Table 4: Ablation Study to Disambiguate Distillation and TRM on Qwen2.5-3B**
> | Method                 | NQ    | TriviaQA | PopQA | HotpotQA | 2wiki | Musique | Bamboogle | Avg.  |
> | ---------------------- | ----- | -------- | ----- | -------- | ----- | ------- | --------- | ----- |
> | Search-R1              | 36.93 | 54.48    | 35.85 | 32.65    | 32.47 | 12.08   | 24.80     | 32.75 |
> | Search-R1-ORM          | 39.47 | 56.17    | 40.93 | 29.63    | 26.65 | 6.83    | 9.60      | 29.90 |
> | Search-R1-TRM-verifier | 35.65 | 54.10    | 36.46 | 33.91    | 33.90 | 13.86   | 27.20     | 33.58 |
> | Search-R1-TRM (ours)   | 39.58 | 57.78    | 40.61 | 34.80    | 33.22 | 12.91   | 25.60     | 34.93 |
>
> [3] Bowen Jin, Hansi Zeng, Zhenrui Yue, Jinsung Yoon, Sercan O Arik, Dong Wang, Hamed Zamani, and Jiawei Han. Search-r1: Training LLMs to reason and leverage search engines with reinforcement learning. In Second Conference on Language Modeling, 2025.

---

> ### Author Response · Authors · 2025-11-23
> **Supplementary Experimental Results (part 4/4)**
>
> ### Details of TRM Training Data ---> Weakness 6 and 9
> **Data construction pipeline** The data construction pipeline, described in Section 2.2 of the original manuscript, consists of two main steps:
> * Rollout collection: Given prompts and available tools, the LLM (DeepSeek-R1 in this work) generates complete trajectories.
> * Tool-call evaluation: For each trajectory, the LLM evaluates each tool call along two dimensions (necessity and quality, binary: 0 or 1). The final score for each tool call is computed as the product of these two dimension scores.
>
> All prompts used for data generation are provided in Appendix A of the original manuscript.
>
> **Data quality** Regarding data quality, we randomly sampled 100 trajectories and evaluated them using both human annotators and a multi-model judge (DeepSeek-R1, Qwen2.5-72B-Instruct, and GPT-4o-mini). As shown in Table 5, the resulting dataset demonstrates relatively high quality. We also plan to open-source all code and data to provide full details and facilitate further research in this area.
>
> Relevant results have been updated in lines 1420–1431 of the revised manuscript.
>
> **Table 5: TRM Training Data Quality Evaluation (Accuracy)**
> | Evaluation Method | Necessity (%) | Quality (%) |
> | ----------------- | ------------- | ----------- |
> | Human             | 89            | 89          |
> | Multi-model       | 86            | 75          |
>
> ---
>
> ### Ablation on the Necessity and Quality of Tool Calls ---> Weakness 9
> We acknowledge this insightful suggestion. Table 6 shows that using quality-only yields the lowest model performance. Table 7 further reveals that relying solely on quality leads to overuse of tools, which negatively impacts performance. In contrast, using necessity-only reduces the number of tool calls but may compromise the quality of each call, also limiting overall performance. Combining both necessity and quality achieves the best overall performance while maintaining a relatively stable number of tool calls across datasets.
>
> Relevant results have been updated in lines 480–485 of the revised manuscript.
>
> **Table 6: Model Performance Ablation on Tool-Call Necessity and Quality (Qwen2.5-3B-Instruct)**
> | Method         | NQ    | TriviaQA | PopQA | HotpotQA | 2wiki | Musique | Bamboogle | Avg.  |
> | -------------- | ----- | -------- | ----- | -------- | ----- | ------- | --------- | ----- |
> | quality-only   | 38.95 | 56.17    | 38.94 | 31.06    | 27.93 | 8.44    | 16.00     | 31.07 |
> | necessity-only | 37.42 | 54.06    | 37.21 | 32.46    | 31.80 | 11.46   | 21.60     | 32.29 |
> | both           | 39.58 | 57.78    | 40.61 | 34.80    | 33.22 | 12.91   | 25.60     | 34.93 |
>
> **Table 7: Tool-Call Number Ablation on Necessity and Quality (Qwen2.5-3B-Instruct)**
> | Method         | NQ   | TriviaQA | PopQA | HotpotQA | 2wiki | Musique | Bamboogle | Avg. |
> | -------------- | ---- | -------- | ----- | -------- | ----- | ------- | --------- | ---- |
> | quality-only   | 3.96 | 3.94     | 3.93  | 3.96     | 3.98  | 3.99    | 3.93      | 3.96 |
> | necessity-only | 2.58 | 2.68     | 2.58  | 2.83     | 2.95  | 3.31    | 2.84      | 2.82 |
> | both           | 2.37 | 2.39     | 2.37  | 2.85     | 3.35  | 3.23    | 2.71      | 2.75 |

---

> ### Author Response · Authors · 2025-11-27
> **Inquiry About Discussion Status**
>
> Dear Reviewer TgrD,
>
> Thank you again for your time and for the constructive comments on our paper. We fully appreciate that everyone has many competing commitments, and we are grateful for your efforts. As the discussion deadline is approaching, we just wanted to briefly follow up. To address your concerns, we have conducted additional experiments and provided further clarifications to the best of our ability.
>
> If you have any remaining questions or would like to discuss any point in more detail, we would be very happy to continue the discussion.
>
> Best regards,
>
> Team 22507

---

### Author Response · Authors · 2025-12-01
**Summary of our work, reviewer feedback, and our responses (part 2/2)**

### Weaknesses and our responses
* Lack of direct comparison with process-supervised tool-use methods (TgrD, S77r)
  > We added experimental comparisons with two representative baselines: StepSearch, a search-specific process-supervised method, and AgentPRM, a general process-supervised method. Across seven search-based QA benchmarks, TRM consistently outperforms both StepSearch and AgentPRM, indicating that our TRM provides more reliable and effective supervision than both generic and hand-crafted process-level signals.
* Separating TRM contributions from LLM distillation effects (TgrD, R2Sf)
  > It is unclear whether the observed performance gains stem from TRM’s per-tool-call reward signals or merely from distillation of strong LLMs. To address this, we conducted an ablation study introducing two baselines trained on the same distillation data: (1) a trajectory-level Outcome Reward Model (ORM), and (2) using TRM as a verifier, where the geometric mean of per-tool-call scores is combined with answer correctness. The results show that using TRM to monitor each tool call consistently yields the best performance.
* Lack of analysis on TRM training data quality (TgrD, 8EFK, S77r, R2Sf)
  > We evaluated 100 randomly sampled TRM training samples using human annotators and a multi-model judge. The dataset shows high quality.
* Lack of ablation on the necessity and quality of tool calls (TgrD, 8EFK, S77r, R2Sf)
  > In TRM training data distillation, each tool call is evaluated along two dimensions: necessity and quality. We conducted ablation experiments on these dimensions. Using quality-only leads to overuse of tools and lower performance, while necessity-only reduces tool-call numbers but may compromise call quality. Combining both necessity and quality yields the best overall performance and maintains a stable number of tool calls across datasets.
* On the generality beyond search and code tools (S77r, R2Sf)
  > We further validated our method in a multi-tool scenario, including tasks such as library management, employee management, and travel planning. Incorporating TRM consistently improves performance, demonstrating its applicability beyond search and code tools.
* Lack of analysis on resource overhead introduced by TRM (TgrD)
  > Incorporating TRM adds only an 8.8% overhead per training step. BoN inference time increases by 50% per sample, but the absolute evaluation time remains small (20 minutes for full BoN), making the additional compute cost practically negligible.
* Lack of Tool-call number comparison in GRPO: turn-level vs. group level (TgrD)
  > Turn-level and group-level advantage estimation in GRPO differ in reward aggregation. Group-level aggregates all tool-call rewards at once, while turn-level divides tool calls sequentially into groups (e.g., first calls form group 1, second calls form group 2, etc.) and computes advantages separately for each group. Group-level generally produces fewer tool calls, reflecting a “fewer calls is better” bias, whereas turn-level encourages more balanced tool usage. We have supplemented the evaluation with tool-call counts on benchmarks, which confirm this trend.
* Effect of TRM Size and Scaling Trend (S77r)
  > We do observe a scaling trend that is broadly consistent with the intuition that larger reward models can provide better supervision. However, we also find that this trend is not purely monotonic. When the amount of distillation data is relatively limited, overly large TRMs are more prone to overfitting, which can in turn hurt downstream performance.
* Limitations in the scope of this Work
  > - *Open-ended task (8EFK, R2Sf)* Our work aims to improve reinforcement learning (RL) for tool-integrated reasoning in settings where the final outcome can be reliably verified. In fully open-ended tasks, where the correctness of the final outcome cannot be robustly determined, RL based purely on outcome rewards is inherently challenging, and is therefore outside the scope of this study.
  > - *Long rollouts* (S77r) In practice, we truncate trajectories to a reasonable length (~8K tokens). Additionally, handling very long rollouts reliably remains an open problem and is outside the scope of this study.
  >
  > ---
  >
  > These limitations have been explicitly updated and discussed in the revised manuscript (lines 835-844).

We sincerely thank all reviewers and committee members for their time and effort. During the discussion period, we have addressed reviewers’ questions and clarified misunderstandings to the best of our ability. We understand that due to the OpenReview bug, reviewers have not yet responded, which may add pressure on the Area Chairs. We provide this summary to help you quickly navigate our work, the reviews, and our detailed responses.

Best regards,

Team 22507

[1] Li, Lei, et al. "Tool-augmented reward modeling." In The Twelfth International Conference on Learning Representations, 2024.

---

### Author Response · Authors · 2025-12-01
**Summary of our work, reviewer feedback, and our responses (part 1/2)**

Dear ACs, SACs, and PCs,

We sincerely thank you all for your time and insightful feedback during the review process. To present our situation as clearly and fairly as possible, we would like to briefly **(1) outline our work** and **(2) summarize the reviews and explain how we have addressed the reviewers' weaknesses point by point in our author response**.

## Outline of our work
> **Problem and Motivation (lines 035-079)** Current methods for training tool-integrated reasoning in LLMs predominantly rely on outcome-only rewards, which provide **undifferentiated supervision to all tool calls** in a trajectory. As a result, even correct tool calls are penalized when the final outcome is incorrect, leading to suboptimal agent performance. Motivated by this limitation, we propose introducing **per-call reward signals** generated by a Tool-call Reward Model (TRM) that assign a separate evaluation to each tool call, providing fine-grained guidance throughout the trajectory.
>
> ---
>
> **Challenges (lines 079-094)** 1) *TRM exploration*: Although TRMs can be viewed as a form of process reward model (PRM), their focus is fundamentally different from traditional PRMs that mainly target reasoning quality on tasks such as mathematical problem solving. In our setting, the TRM must explicitly evaluate individual tool calls within a trajectory, **a direction that has received little attention so far**. As a result, key design questions remain underexplored, including what model scale is sufficient for TRMs, how much training data they require, and how training data should be constructed. 2) *TRM exploitation:* A further challenge is **how to integrate these per-call reward signals into standard RL algorithms such as PPO and GRPO**, so that fine-grained supervision from TRMs can be translated into consistent performance improvements for tool-using agents.
>
> ---
>
> **Methods** For the first challenge, we develop a workflow to distill training data from frontier LLMs and systematically study how TRM performance scales with data volume and model size (**lines 113-161**). For the second challenge, we refine the credit assignment strategy by allocating tool-call rewards to the end of each tool invocation in PPO, and we introduce turn-level advantage estimation in GRPO to better exploit per-call supervision from TRMs (**lines 183-250**).
>
> ---
>
> **Conclusion** 1) Under our proposed workflow, a 3B TRM trained on only 10k distilled trajectories already achieves strong performance, indicating that effective process-level supervision can be obtained with moderate model size and data scale (**lines 262-308**). 2) Across two representative tool-use scenarios (search-based QA and Python-based mathematical problem solving), we observe consistent performance improvements when incorporating TRM-based per-call evaluation of tool use, for multiple model sizes including 1.5B, 3B, and 7B (**lines 311-404**).

For each part above, we also provide the corresponding line numbers in our manuscript for your convenience. We kindly refer you to the manuscript for further details.

## Summary of reviews and our responses
### Serious concern about the reliability of one review
While we received two clearly positive reviews (score 6) and one moderate review (score 4, with “good” (score 3) in all sub-aspects including soundness, presentation, and contribution), the score-2 review is largely driven by **major factual misunderstandings of our work**.

Specifically, this review (1) conflates our tool-call reward model with prior work on tool-augmented reward modeling [1], which is **extremely different both conceptually and operationally**, (2) misinterprets the motivation of TRMs and their relationship to process reward models (PRMs), and (3) misunderstands our evaluation protocol for TRMs. We have addressed each of these points with clear and detailed clarifications in our author response (**see Reviewer TgrD, *Clarification of Certain Misunderstandings (part 1/4)***).

### Strengths highlighted by the reviewers
* Clear and timely motivation, tackling the limitations of outcome-only rewards by providing per-call supervision that enhances tool-integrated reasoning (TgrD, 8EFK, R2Sf)
* Extensive experiments showing consistent improvements on search-based QA and code tasks (8EFK, S77r, R2Sf)
* Facilitates systematic exploration of TRMs, providing insights into model size and training data volume (TgrD)
* Clear and well-written presentation, with thorough and detailed explanations (8EFK, S77r)

---

### Meta-Review · Area_Chair_y6TL · 2025-12-22

**Summary:**

The reviewers acknowledged the paper's timely motivation: addressing the limitations of outcome-only rewards in RL for tool-using LLMs by proposing a fine-grained Tool-call Reward Model (TRM). Strengths included clear writing and extensive experiments. However, significant concerns were raised. Key criticisms involved insufficient comparison with existing process-supervised methods, potential confounding factors from using a strong LLM for TRM data distillation, and a perceived narrow scope limited primarily to search and code tools.

**Reviewer Concerns:**

The author rebuttal effectively addressed several major concerns. They provided direct comparisons with process-supervised baselines (StepSearch, AgentPRM). Ablation studies clearly demonstrated the individual and combined importance of "necessity" and "quality." They reported data quality metrics from human and multi-model evaluation and quantified resource overhead, showing it is manageable. To demonstrate generality, they added experiments in diverse multi-tool scenarios beyond search and code.

While the rebuttal strengthened the paper, some concerns remain as acknowledged limitations. The core methodology still relies on distillation from a strong LLM, and while ablations disentangled TRM's contribution, the fundamental dependence on this data generation process is a conceptual limitation. The applicability to fully open-ended tasks, where final outcomes are unverifiable, remains outside the paper's scope.

**Reviewer Scores:**

- Reviewer TgrD (Score: 2): The authors' detailed clarifications and substantial new experiments directly countered reviewer's criticisms. It is likely the score would have increased.
- Reviewer 8EFK (Score: 6): Their primary concerns (data quality, open-ended tasks) were directly addressed with new quality evaluations and a clear statement of scope. This reviewer would likely maintain the score
- Reviewer S77r (Score: 6): This reviewer's specific questions (ablations on necessity/quality, comparison to PRM baselines, generality) were answered with new results. This reviewer would likely maintain the score
- Reviewer R2Sf (Score: 4): Their concerns about generality and distillation were mitigated by the new multi-tool experiments and ablation studies. This reviewer would likely maintain the score

---

### Decision · Program_Chairs · 2026-01-26

Accept (Poster)